# Translation of zinc finger domains induces ribosome collision and Znf598-dependent mRNA decay in zebrafish

Kota Ishibashi[1], Yuichi Shichino[2], Peixun Han[2¤], Kimi Wakabayashi[1], Mari Mito[2], Toshifumi Inada[3], Seisuke Kimura[4,5], Shintaro Iwasaki[2,6], Yuichiro Mishima[1,2]*

1 Department of Frontier Life Sciences, Faculty of Life Sciences, Kyoto Sangyo University, Kyoto, Japan, 2 RNA Systems Biochemistry Laboratory, RIKEN Cluster for Pioneering Research, Saitama, Japan, 3 Institute of Medical Science, The University of Tokyo, Tokyo, Japan, 4 Department of Industrial Life Sciences, Faculty of Life Sciences, Kyoto Sangyo University, Kyoto, Japan, 5 Center for Plant Sciences, Kyoto Sangyo University, Kyoto, Japan, 6 Department of Computational Biology and Medical Sciences, Graduate School of Frontier Sciences, The University of Tokyo, Chiba, Japan

¤ Current address: Center for Frontier Research, National Institute of Genetics, Shizuoka, Japan
* mishima@cc.kyoto-su.ac.jp

**Data Availability Statement:** All relevant data are within the paper and its Supporting information files, except for the sequence data, which are available at GEO. The accession numbers for the

## Abstract

Quality control of translation is crucial for maintaining cellular and organismal homeostasis. Obstacles in translation elongation induce ribosome collision, which is monitored by multiple sensor mechanisms in eukaryotes. The E3 ubiquitin ligase Znf598 recognizes collided ribosomes, triggering ribosome-associated quality control (RQC) to rescue stalled ribosomes and no-go decay (NGD) to degrade stall-prone mRNAs. However, the impact of RQC and NGD on maintaining the translational homeostasis of endogenous mRNAs has remained unclear. In this study, we investigated the endogenous substrate mRNAs of NGD during the maternal-to-zygotic transition (MZT) of zebrafish development. RNA-Seq analysis of zebrafish *znf598* mutant embryos revealed that Znf598 down-regulates mRNAs encoding the C2H2-type zinc finger domain (C2H2-ZF) during the MZT. Reporter assays and disome profiling indicated that ribosomes stall and collide while translating tandem C2H2-ZFs, leading to mRNA degradation by Znf598. Our results suggest that NGD maintains the quality of the translatome by mitigating the risk of ribosome collision at the abundantly present C2H2-ZF sequences in the vertebrate genome.

## Introduction

The translation of mRNA into protein is a fundamental and a costly process in the cell [1]. To minimize the cost of translation and maintain proteome integrity, cells harness multiple quality control mechanisms that prevent faulty translation. The ribosome lies at the core of such translational quality control mechanisms; aberrant positioning, movement, and structure of ribosomes during mRNA translation serve as cues to trigger specific quality control pathways [2,3]. The failure of translational quality control pathways results in the production of harmful proteins and protein aggregates, leading to pathogenic outcomes [4–6].

sequence data reported in this paper are as follows. GSE236143 (RNA-Seq in zebrafish), GSE236144 (additional disome profiling in zebrafish), and GSE233374 (BRIC-Seq). Custom codes used in this study are available at Zenodo (DOI: 10.5281/zenodo.13864507).

**Funding:** This work was supported by the Japan Society for the Promotion of Science (JSPS) (JP23H02413 and JP22K19300 to YM and JP22K15042 to KI) and the Japan Agency for Medical Research and Development (AMED) (AMED PRIME, JP20gm6310017) to YM. SI was supported by JSPS (JP23H02415 and JP23H00095), MEXT (JP24H02307), AMED (AMED-CREST, JP20gm1410001), and RIKEN (Pioneering Projects "Biology of Intracellular Environments"). YS was supported by JSPS (JP21K15023 and JP23K05648), MEXT (JP21H05734, 23H04268), and RIKEN (Pioneering Projects "Biology of Intracellular Environments"). SK was supported by JSPS (JP21H02513). PH was supported by the JSPS Research Fellowship for Young Scientists (JP22J13110). The funders had no role in study design, data collection and analysis, decision to publish, or preparation of the manuscript.

**Competing interests:** The authors have declared that no competing interests exist.

**Abbreviations:** dpf, days postfertilization; hpf, hours postfertilization; MO, morpholino oligonucleotide; MZT, maternal-to-zygotic transition; NGD, no-go decay; RQC, ribosome-associated quality control; ZGA, zygotic genome activation.

Persistent stalling of the ribosome during the translation elongation process is detrimental to cells, as it perturbs ribosome traffic on mRNA and exposes the premature polypeptide to the intracellular environment. Ribosomes stall when encountering multiple translational challenges, including tRNA shortages, decoding defects, obstruction by RNA secondary structures, chemical damage to RNA, inhibitory nascent peptide sequences, and combinations thereof [7–9]. Eventually, the trailing ribosome catches up and collides with the leading stalled ribosome, forming an unusual ribosome dimer structure called a disome (or trisome if 3 ribosomes collide) [10–13]. The cellular machinery recognizes disomes in multiple ways to solve or mitigate translational problems [14,15].

In eukaryotes, the faulty translation leading to disome formation is terminated by splitting the stalled ribosome and degrading the premature nascent polypeptide by ribosome-associated quality control (RQC) [16–18]. In addition, no-go decay (NGD) degrades the mRNA on which ribosomes stall, preventing the accumulation of stall-prone mRNAs [19–21]. Although the coordination and order of RQC and NGD require further investigation, ubiquitination of the stalled ribosome in the disome by ZNF598 (Hel2 in yeast) is the most upstream step in both pathways [10,12,22–25]. ZNF598/Hel2 ubiquitinates the ribosomal small subunit proteins RPS10/eS10 and RPS20/uS10 at conserved lysine residues, followed by splitting of the stalled ribosome by the RQC-trigger (RQT) complex (also known as the ASC-1 complex; ASCC) [10,24,26–29]. Ubiquitination of the stalled ribosome by ZNF598/Hel2 also induces NGD in yeast and *Caenorhabditis elegans* [11,30,31], partly via the function of the ubiquitin-binding endonuclease Cue2/NONU-1 [31–33]. Alternatively, stall-prone mRNAs might be degraded via GIGYF family proteins that bind to ZNF598/Hel2 [34–36]. Whether vertebrate NGD occurs coincidently with ribosome collision and RQC remains controversial [25,37], but Znf598 is required for stall-induced decay of a reporter mRNA in zebrafish embryos [38]. These observations suggest the presence of a conserved mechanism acting on RQC and NGD in eukaryotes.

While the molecular mechanism of NGD has been revealed using defined stall-inducing sequences [11,20,31–33,37,39], the endogenous substrate mRNAs of NGD have not been fully characterized. Chemically damaged mRNAs induce ribosome stalling and are potential NGD substrates; however, endogenous mRNAs that are constitutively degraded via NGD under normal conditions remain unclear [8,40]. Given the widespread distribution of disomes in the transcriptome, it is reasonable to speculate that NGD has a broad impact on endogenous gene expression [41–44]. However, not all disome-forming mRNAs are stabilized in the absence of Hel2 in yeast, making it challenging to identify endogenous mRNAs that are subject to NGD [43]. As a result, only a handful of endogenous NGD target mRNAs have been identified thus far [31,35,37]. Comprehensive identification of the endogenous NGD target mRNAs is needed to understand the physiological role of NGD besides its established role in suppressing transiently generated aberrant mRNAs.

Early embryogenesis is a unique model system for studying mRNA decay in vivo. The massive decay of maternal mRNA and concurrent zygotic genome activation (ZGA) constitute a conserved developmental transition in metazoans called the maternal-to-zygotic transition (MZT) [45–48]. The molecular mechanisms underlying MZT have been extensively characterized in zebrafish, in which codon-mediated decay and zygotically transcribed miR-430 play pivotal roles in maternal mRNA clearance in combination with m⁶A modification [49–54]. By combining a zebrafish mutant of *znf598* and a reporter mRNA containing a ribosome stall sequence, we previously showed that ribosome stalling induces mRNA decay in a Znf598-dependent manner, demonstrating that NGD is active during the MZT [38]. Codon-mediated decay was unaffected by the loss of Znf598, indicating that NGD is mechanistically distinct from codon-mediated decay [38]. However, the contribution of NGD to shaping the mRNA expression patterns during the MZT has yet to be explored.

In this study, we investigated the endogenous substrate mRNAs of NGD during the MZT using *znf598* mutant zebrafish embryos. Our RNA-Seq analysis showed that Znf598 down-regulates mRNAs encoding the C2H2-type zinc finger domain (C2H2-ZF), many of which are expressed zygotically at the onset of MZT. We combined disome profiling and reporter analysis to demonstrate that C2H2-ZF sequences induce ribosome stalling and Znf598-dependent mRNA decay. Based on these results, we propose that C2H2-ZF sequences are endogenous ribosome stallers in vertebrates, and the stalled ribosomes induce NGD in zebrafish embryos.

## Results

### The physiological impact of the loss of Znf598 in zebrafish

A zebrafish *znf598* mutant strain with an 11-base deletion in exon 1, predicted to cause frame-shifting at the middle of the RING domain, was generated previously [38]. Embryos lacking maternal and zygotic *znf598* activity (MZ*znf598*) are defective in mRNA decay induced by the ribosome stall sequence of hCMV *gp48* uORF2 and cannot ubiquitinate Rps10/eS10 in response to translational stresses [38,55].

To ascertain the physiological impact of the loss of Znf598, we analyzed the viability and growth of the *znf598* mutant strain. Fish obtained by crossing heterozygous *znf598* mutants showed a moderately skewed ratio of genotypes in the tenth week after birth (S1A Fig). The difference in the body length between the genotypes was evident, suggesting that the loss of zygotic *znf598* activity reduced the growth rate (S1B Fig). We then crossed *znf598* homozygous males and females to obtain MZ*znf598* embryos. MZ*znf598* embryos were morphologically indistinguishable from wild-type embryos during the gastrulation and prehatching stages (Fig 1A). Consistent with the previous studies in yeast and human cultured cells [56,57], phosphorylation of eIF2α was elevated in MZ*znf598* embryos (Fig 1B). This result implies the accumulation of collided ribosomes and induction of integrated stress response in MZ*znf598* embryos. MZ*znf598* larvae became shorter than wild-type larvae at 5- and 9-days postfertilization (dpf), suggesting the requirement of Znf598 for supporting posthatching larval growth (Fig 1C and 1D). To further evaluate the growth and survival rates of MZ*znf598*, we cultured the same number of wild-type and MZ*znf598* larvae together in a single tank (S1C Fig). Consistent with the analysis of zygotic *znf598* mutant fish, fewer MZ*znf598* fish survived than wild-type fish, and the body length of the survived fish were shorter than those of the wild-type fish at 12 weeks after birth (S1D Fig). MZ*znf598* fish apparently caught up with wild-type fish in the body length by 28 weeks after birth without a reduction in the survival rate (S1D and S1E Fig). These results indicate that Znf598 is not obligatory for morphogenesis during embryogenesis but is required for suppressing the accumulation of collided ribosomes and supporting post-embryonic growth and survival in zebrafish.

### Transcriptome analysis of MZ*znf598* during the MZT

In theory, mRNAs that are degraded via NGD should accumulate in the absence of Znf598. Since NGD reporter mRNA is degraded in a Znf598-dependent manner during the MZT [38], we performed RNA-Seq analysis before (1-hour postfertilization: 1 hpf) and after (6 hpf) the MZT with wild-type and MZ*znf598* embryos to survey endogenous NGD target mRNAs (Fig 2A). Analysis of the RNA-Seq data revealed mRNAs that were significantly up-regulated or down-regulated at each stage (FDR < 0.01) (Fig 2B). Functional enrichment analysis revealed that no specific term was overrepresented in genes whose expression changed at 1 hpf. In contrast, 3 specific terms were overrepresented among the genes whose expression changed at 6 hpf (Fig 2C). Notably, 2 terms related to C2H2-type zinc finger domains (C2H2-ZF) were enriched in genes up-regulated at 6 hpf.

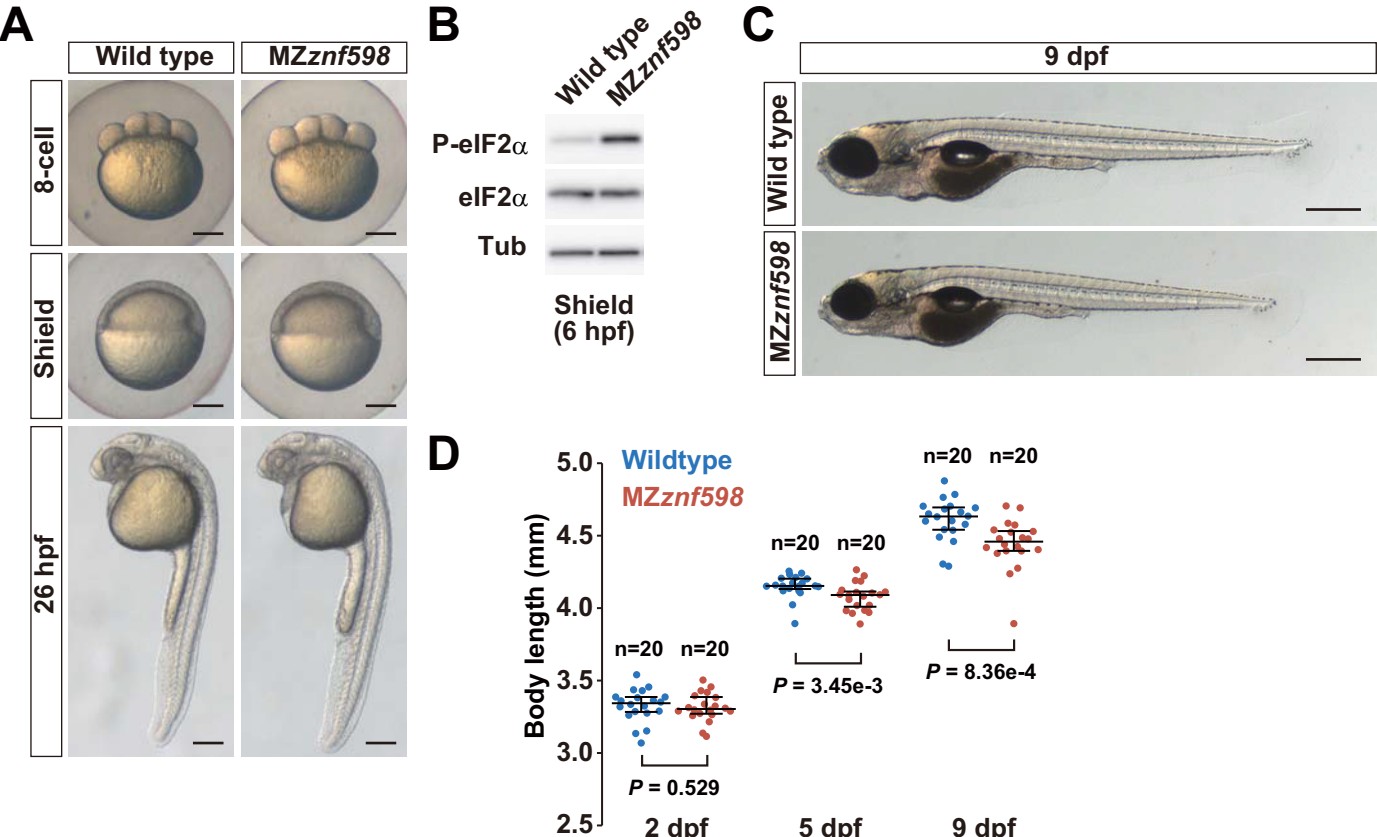

**Fig 1. Analysis of zebrafish MZ*znf598* embryos and larvae.** (A) Brightfield images of wild-type and MZ*znf598* embryos at the 8-cell stage, shield stage, and 26 hpf. The scale bars indicate 200 μm. (B) Western blotting of phosphorylated eIF2α in wild-type and MZ*znf598* zebrafish embryos at 6 hpf. Total eIF2α and Tubulin (Tub) were detected as loading controls. (C) Brightfield images of wild-type and MZ*znf598* embryos at 9 dpf. Scale bars indicate 500 μm. (D) Distributions of body length in wild-type and MZ*znf598* fish at 2, 5, and 9 dpf. *P* values were calculated by the Mann–Whitney U test (two-tailed). The numerical data underlying this figure can be found in S1 Data. dpf, days postfertilization; hpf, hours postfertilization.

To confirm the changes in the mRNA expression of the C2H2-ZF genes in MZ*znf598*, we identified genes encoding C2H2-ZF based on PROSITE annotation and analyzed their expression at 6 hpf. Among the 15,245 genes expressed at 6 hpf (>1 count per million mapped reads), 882 genes encoded C2H2-ZF genes. Of those, 57 C2H2-ZF genes were up-regulated and 18 genes were down-regulated in MZ*znf598* at 6 hpf (FDR < 0.01) (Fig 2D). This biased distribution resulted in a significant enrichment of C2H2-ZF genes among the genes up-regulated in the MZ*znf598* embryos (57 out of 247 protein-coding genes, 23.1%) compared to all the genes expressed at 6 hpf (882 C2H2-ZF genes out of 15,245 protein-coding genes, 5.79%) (Fig 2E and S1 Table). Consistent with this bias, 61.6% of the C2H2-ZF genes exhibited positive fold changes in expression in MZ*znf598* embryos (543 out of 882 C2H2-ZF genes expressed) (Fig 2F). These trends were not observed with another zinc finger domain, RING-type ZF (RING-ZF) (Fig 2E and 2F), suggesting a specific impact of Znf598 deficiency on the expression level of C2H2-ZF genes.

To determine the expression patterns of the genes up-regulated in MZ*znf598*, we analyzed changes in the mRNA level of these genes during the MZT by comparing RNA-Seq data from wild-type embryos at 1 hpf and 6 hpf. In accordance with the method of the previous study [50], we considered genes whose mRNA level decreased or increased more than 2-fold from 1 hpf to 6 hpf to be predominantly maternal or zygotic genes, respectively. Other genes were

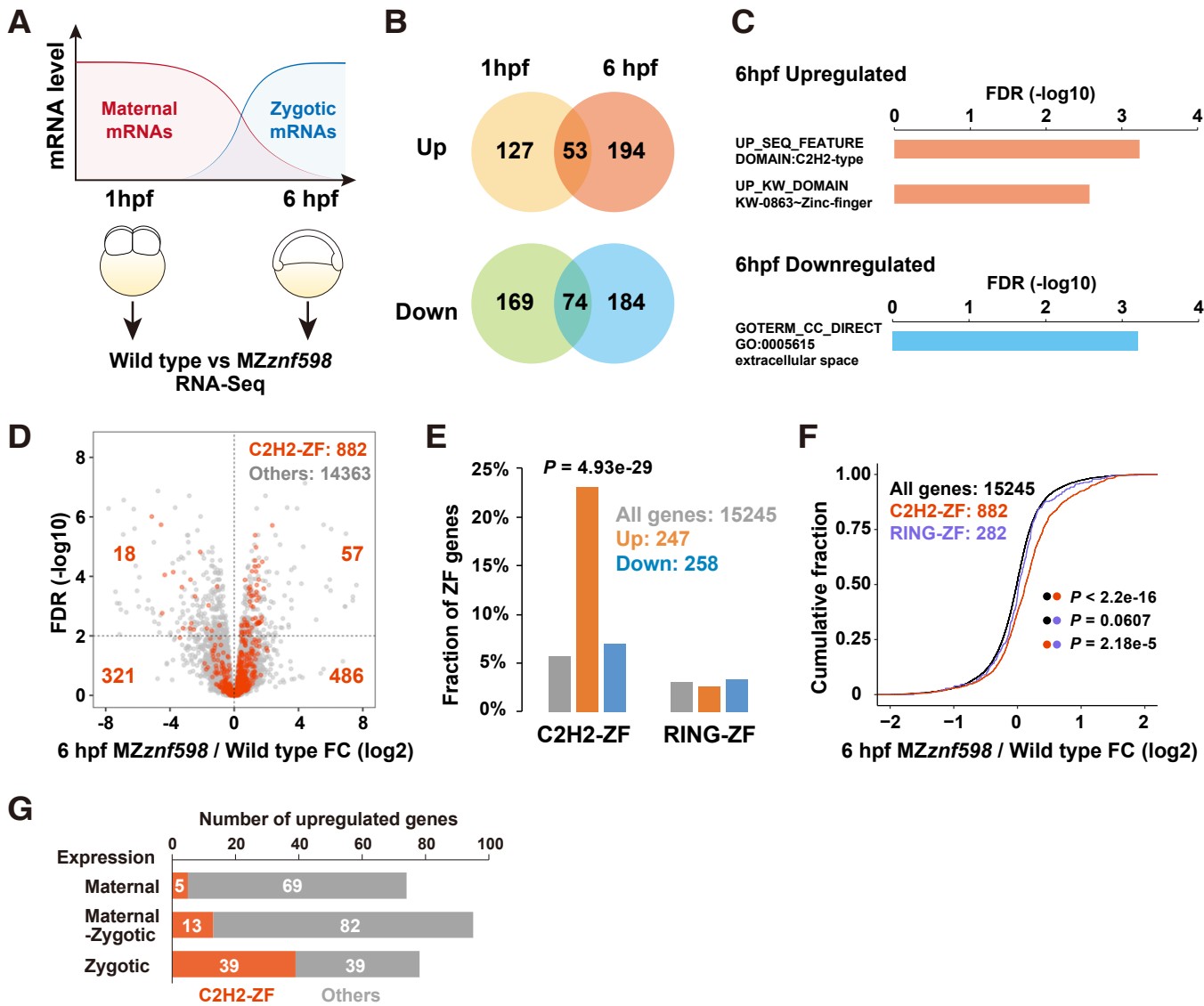

**Fig 2. Transcriptome analysis of MZ*znf598* embryos during the MZT.** (A) A scheme of RNA-Seq analysis comparing wild-type and MZ*znf598* embryos during the MZT. (B) Venn diagrams of mRNAs up-regulated or down-regulated at 1 hpf or 6 hpf (FDR < 0.01). The numbers of genes in each category are shown. (C) Functional enrichment analysis of biological terms associated with genes up-regulated (orange) or down-regulated (light blue) at 6 hpf. Terms that were significantly enriched (FDR < 0.01) are shown. (D) A volcano plot showing the fold change in mRNA expression between wild-type and MZ*znf598* embryos at 6 hpf (x-axis) and FDR (y-axis). Genes encoding C2H2-ZF (red) and other genes (gray) are shown. (E) Bar graph showing the fractions of C2H2-ZF or RING-ZF genes among all the genes (gray), genes up-regulated in MZ*znf598* (orange), and genes down-regulated in MZ*znf598* (blue) at 6 hpf. The *p* value was calculated by the chi-square test. (F) Cumulative distributions of fold changes in mRNA levels between MZ*znf598* embryos and wild-type embryos at 6 hpf. All genes (black), C2H2-ZF genes (red), and RING-ZF genes (purple) are shown. The x-axis shows the fold change, and the y-axis shows the cumulative fraction. The *p* values are shown on the right (Kolmogorov–Smirnov test). (G) Bar graphs summarizing the expression patterns of C2H2-ZF genes (red) and other genes (gray) up-regulated in MZ*znf598* embryos at 6 hpf. The numerical data underlying this figure can be found in S1 Data. hpf, hours postfertilization; MZT, maternal-to-zygotic transition.

considered maternal-zygotic genes. The numbers of up-regulated genes were comparable among the 3 classes, whereas up-regulated C2H2-ZF genes showed a strong bias toward zygotic genes (Fig 2G). These results suggest that the NGD regulates a subset of mRNAs expressed during the MZT and that mRNAs encoding C2H2-ZF are enriched in these mRNAs.

## Characterization of maternal NGD target mRNAs

To characterize 74 maternal mRNAs up-regulated in MZ*znf598* (maternal NGD target candidates), we compared these mRNAs with previously defined unstable maternal mRNAs degraded during the MZT. In zebrafish, maternal mRNAs with low codon optimality are degraded via the maternal decay pathway, while mRNAs whose degradation depends on zygotic factors, as exemplified by miR-430, are categorized in the zygotic pathway [49–51]. We found that maternal mRNAs up-regulated in MZ*znf598* were distinct from maternal mRNAs degraded by maternal and zygotic decay pathways (S2A Fig). Conversely, mRNA degradation via maternal or zygotic decay pathways was unaffected by the loss of Znf598 (S2B and S2C Fig). Similarly, the stability of maternal mRNAs containing m⁶A modifications, which act in concert with codon optimality and miR-430 to clear maternal mRNAs [52–54], was unchanged in the MZ*znf598* embryos (S2D Fig). These observations support the view that NGD is mechanistically distinct from mRNA decay pathways mediated by codon optimality, miR-430, and m⁶A modification.

We selected 2 C2H2-ZF genes (*znf236* and *znf970*) and 3 non-C2H2-ZF genes (*ap3s2*, *mapk14b*, and *piezo2*) from the list of maternal NGD target candidates and validated the RNA-Seq data by qRT-PCR (Fig 3A and 3B). For this validation, we performed 2 rescue experiments; MZ*znf598* embryos were injected with mRNAs encoding Myc-tagged full-length Znf598 or a Znf598 mutant lacking the RING domain essential for ribosome ubiquitination and NGD [11,38]. Among the 5 candidate mRNAs tested, *znf236* and *znf970* were decreased from 2 hpf to 6 hpf in a Znf598- and its RING domain-dependent manner (Fig 3A). Time course qRT-PCR analysis further confirmed that the expression of the *znf236* and *znf970* mRNAs decreased during MZT in wild-type embryos, and this reduction was suppressed in MZ*znf598* embryos (Fig 3C). In contrast, the *ap3s2*, *mapk14b*, and *piezo2* mRNAs accumulated at higher levels in the MZ*znf598* embryos than in the wild-type embryos, as observed in the RNA-Seq data, but were not down-regulated by the full-length Znf598 rescue (Fig 3A). These mRNAs might be up-regulated indirectly due to the loss of Znf598 in previous stages (e.g., oogenesis). Notably, both *znf236* and *znf970* encoded nearly 30 copies of tandem C2H2-ZFs (Fig 3D).

We further focused on *znf236* mRNA to determine whether its degradation pattern matched the characteristics of NGD. First, we blocked the translation of *znf236* mRNA by injecting a specific morpholino oligonucleotide (MO) that masked the initiation codon from the ribosome [58]. *znf236* mRNA was stabilized in the presence of a specific MO, demonstrating that its degradation required active translation, similar to that of NGD (Fig 3E) [21]. Second, we performed a poly(A) test (PAT) assay for *znf236* mRNA in wild-type and MZ*znf598* embryos at 6 hpf (Fig 3F and 3G) [50,59]. *znf236* mRNA was not subject to active deadenylation as observed with the NGD reporter mRNA [38]. Third, we analyzed *znf236* mRNA in maternal-zygotic Rps10 K139R/K140R mutant (MZ*rps10*KR) embryos, which lack Znf598-mediated ribosome ubiquitination sites on Rps10/eS10 [12,25,55]. We observed up-regulation of *znf236* mRNA in MZ*rps10*KR embryos (S2E Fig). The modest effect of Rps10 ubiquitination site mutations on the *znf236* mRNA stability compared to the loss of Znf598 was presumably due to the presence of other ubiquitination targets for Znf598, such as Rps20/uS10 and Rack1/Asc1 [11,22–24,29,30]. Overall, these data indicate that maternal C2H2-ZF mRNAs are degraded by a Znf598-dependent mechanism after fertilization.

## Characterization of zygotic NGD target mRNAs

Having confirmed the Znf598-dependent decay of maternal C2H2-ZF mRNAs, we next investigated zygotic C2H2-ZF mRNAs up-regulated in MZ*znf598* embryos. Many C2H2-ZF genes

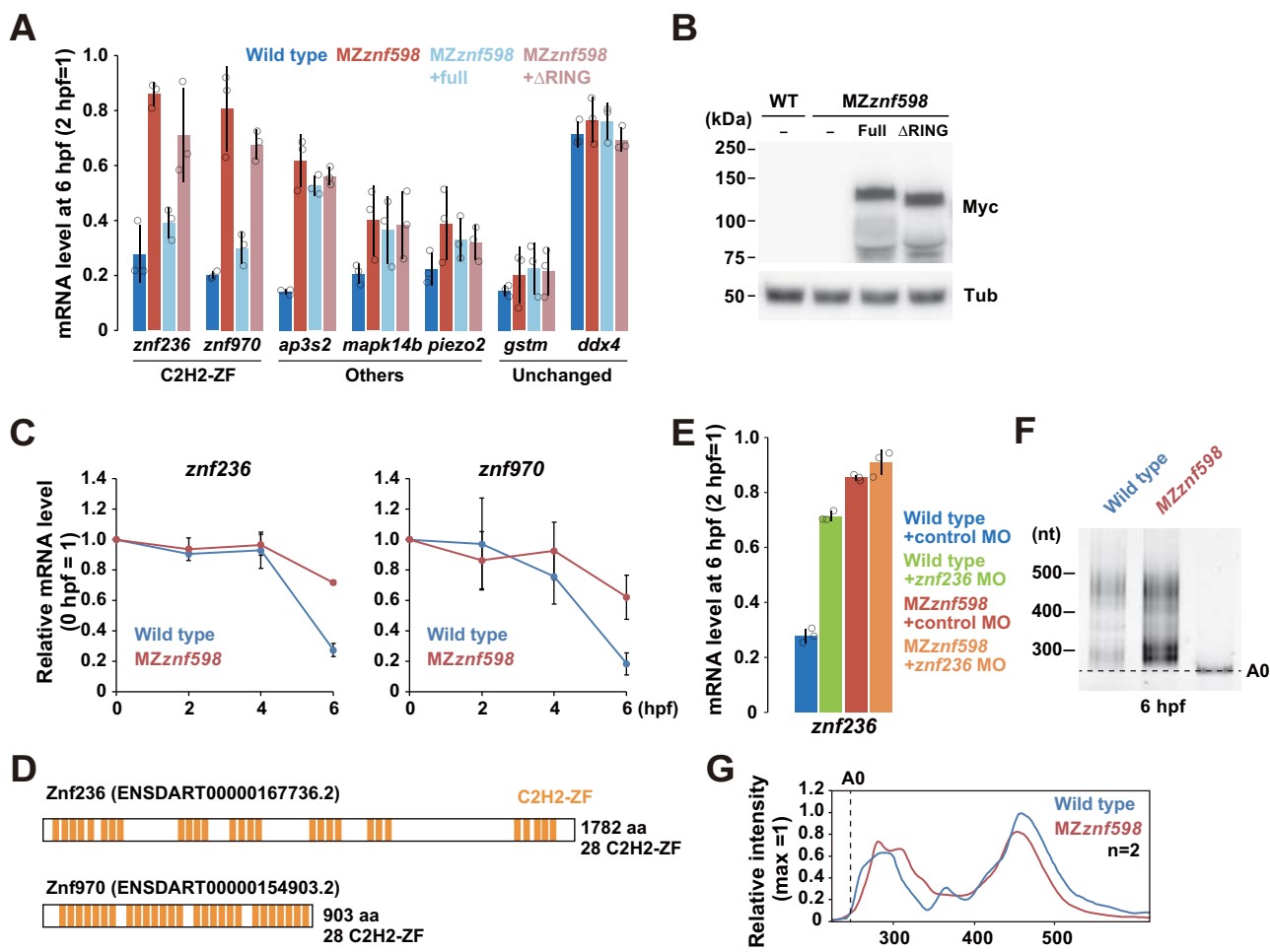

**Fig 3. Validation of maternal mRNAs up-regulated in MZ*znf598*.** (A) qRT-PCR analysis of maternal NGD target candidate mRNAs in wild-type (blue) and MZ*znf598* embryos (red) at 6 hpf relative to 2 hpf. The results for MZ*znf598* embryos rescued by injecting mRNAs encoding Myc-tagged full-length Znf598 (light blue) or a mutant Znf598 lacking the RING domain (pink) are also shown. *gstm* mRNA (miR-430 target) and *ddx4* mRNA (stable maternal mRNA) are shown as controls for Znf598-independent maternal mRNAs. (B) Western blotting to detect Myc-tagged Znf598 proteins at 6 hpf. Tubulin (Tub) was detected as a loading control. (C) Time course qRT-PCR analysis of *znf236* and *znf970* mRNAs in wild-type (blue) and MZ*znf598* (red) embryos. (D) Schematics of the Znf236 and Znf970 proteins. C2H2-ZF is indicated in orange. (E) qRT-PCR analysis of *znf236* mRNA in wild-type and MZ*znf598* embryos at 6 hpf injected with control MO (blue and red) or translation-blocking MO (green and orange). (F) PAT assay of *znf236* mRNA in wild-type and MZ*znf598* embryos at 6 hpf. The lane labeled A0 shows the 3′ UTR fragment without a poly(A) tail. (G) Quantification of the PAT assay in (F). The average values of the 2 experiments are shown. The graphs represent the average of 3 independent experiments in A, C, and E. The error bars show the standard deviation. The open circles in A and E show each data point. The numerical data underlying this figure can be found in S1 Data. hpf, hours postfertilization; MO, morpholino oligonucleotide; NGD, no-go decay.

encode tandem arrays of C2H2-ZFs, with a median of 8 C2H2-ZFs per ORF in zebrafish. The number of tandem C2H2-ZF genes often increases via local gene duplication [60], and approximately one-third of C2H2-ZF genes are present on chromosome 4 in zebrafish (401/1115 C2H2-ZF genes in our data sets) [61,62]. A subset of these C2H2-ZF genes shares a characteristic promoter signature with the zygotic miR-430 cluster and are transcribed earlier at the ZGA during the MZT, followed by transcription of the remaining C2H2-ZF genes [63,64]. Consistent with these studies, mRNAs of C2H2-ZF genes on chromosome 4 were globally up-regulated from 1 hpf to 6 hpf according to our wild-type RNA-Seq data (S3A Fig). Moreover, C2H2-ZF genes on chromosome 4 were significantly up-regulated in MZ*znf598* embryos at 6 hpf, highlighting the pronounced impact of Znf598 on C2H2-ZF genes on chromosome 4 (Figs 4A and S3B). We selected 3 C2H2-ZF genes located on chromosome 4 (*znf1093*, *si:ch73-*

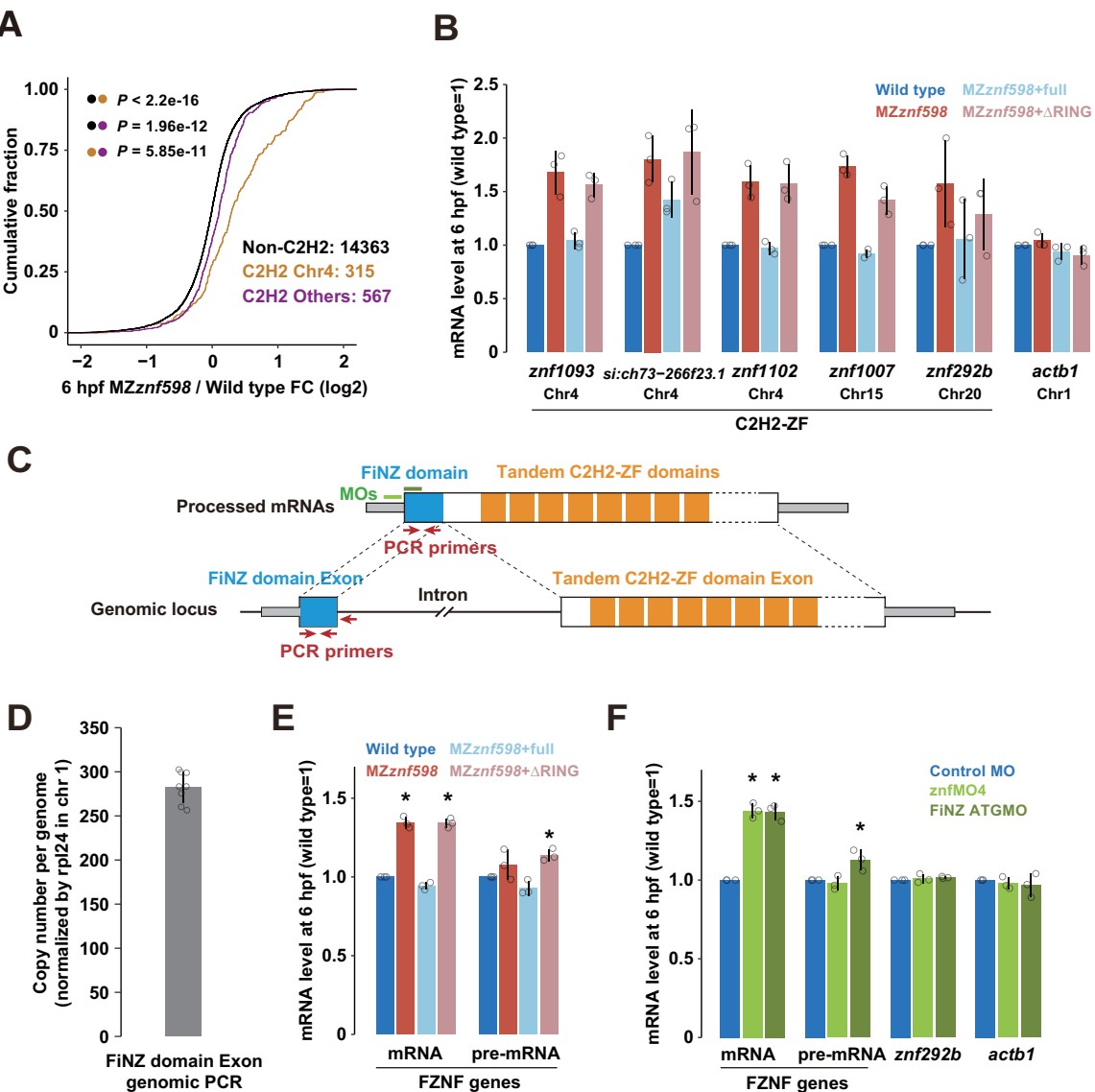

**Fig 4. Validation of zygotic C2H2-ZF mRNAs up-regulated in MZ*znf598*.** (A) Cumulative distributions of fold changes in mRNA levels in MZ*znf598* embryos compared to wild-type embryos at 6 hpf. C2H2-ZF genes on chromosome 4 (orange), C2H2-ZF genes on other chromosomes (purple), and genes without C2H2-ZF (black) are shown. The x-axis shows the fold change, and the y-axis shows the cumulative fraction. The *p* values are shown on the left (Kolmogorov–Smirnov test). (B) qRT-PCR analysis of C2H2-ZF mRNAs in wild-type (blue) and MZ*znf598* embryos (red) at 6 hpf. The results for MZ*znf598* embryos rescued by injecting mRNAs encoding Myc-tagged full-length Znf598 (light blue) or a mutant Znf598 lacking the RING domain (pink) are also shown. *actb1* mRNA is shown as a control for Znf598-independent mRNA. The chromosomal locations are indicated below the gene name. (C) A scheme of typical FZNF mRNA and corresponding pre-mRNA/genetic loci. (D) Genomic qPCR analysis using primers that amplify exons encoding the FiNZ domain. The estimated copy number of FiNZ exons per haploid genome normalized to that of *rpl24* is shown on the y-axis. (E) qRT-PCR analysis of FZNF mRNAs and pre-mRNAs at 6 hpf. The colors are the same as in (B). (F) qRT-PCR analysis of FZNF mRNAs at 6 hpf in wild-type embryos injected with control MO (blue), znfMO4 (light green), or FiNZ ATGMO (green). *znf292b* (a C2H2-ZF gene without the FiNZ domain) and *actb1* (a gene without C2H2-ZF) are shown as controls. The graphs in B, E, and F represent the average of 3 independent experiments. The graphs in D represent the average of 8 independent embryos. The error bars show the standard deviation. The open circles in B, D, E, and F show each data point. Asterisks in E and F indicate *p* < 0.05 (Dunnett's test, compared to wild type or control MO). The numerical data underlying this figure can be found in S1 Data. hpf, hours postfertilization; MO, morpholino oligonucleotide.

*266f23.1*, and *znf1102*) from the list of zygotic NGD target candidates and 2 C2H2-ZF genes on other chromosomes from maternal-zygotic NGD target candidates (*znf1007* and *znf292b*) to validate their Znf598-dependent expression changes by qRT-PCR. All these C2H2-ZF genes were up-regulated in MZ*znf598* and suppressed by Znf598 in a RING domain-dependent manner at 6 hpf, indicating the specific down-regulation of zygotic C2H2-ZF mRNAs by Znf598 (Fig 4B).

The predominant impact of Znf598 on zygotic C2H2-ZF genes on chromosome 4 prompted us to further investigate the common characteristics of these genes. A recent study identified zygotic C2H2-ZF mRNAs on zebrafish chromosome 4 as cyprinid fish-specific tandem C2H2-ZF genes based on their common gene structure and sequence [65]. The first coding exon of these C2H2-ZF genes typically encodes a fish N-terminal zinc-finger associated (FiNZ) domain, and the second coding exon encodes tandem C2H2-ZFs (Fig 4C). These genes were termed FiNZ-ZNF (FZNF), and 684 genes in the zebrafish genome were predicted to be FZNF [65]. Exploiting the sequence similarity of FZNFs, we designed PCR primers that could amplify multiple FiNZ exons in a single PCR reaction (Fig 4C). Genomic qPCR analysis revealed that this primer set amplifies approximately 280 copies of FZNF loci (Fig 4D). This experimental setup allows us to measure the abundance of FZNF mRNAs though qRT-PCR. We observed that FZNF mRNAs were up-regulated in MZ*znf598* embryos and suppressed by Znf598 in a RING domain-dependent manner at 6 hpf (Fig 4E).

While the up-regulation of FZNF mRNAs in the MZ*znf598* embryos suggested mRNA decay by Znf598, transcriptional activation could also explain the observed changes in expression. To address this possibility, we quantified pre-mRNA levels of FZNF genes using primers that amplify the exon–intron boundary (Fig 4C). We observed that the alterations in pre-mRNA levels were limited compared to those in the mature mRNA levels (Fig 4E). RNA-Seq data further supported this conclusion; reads mapped to intron regions of C2H2-ZF genes showed only marginal up-regulation (S3C Fig). Even when we subtracted changes in the intron level, we still observed substantial up-regulation of the mature mRNA level of C2H2-ZF genes in MZ*znf598* (S3D Fig).

Next, we examined whether the FZNF mRNAs were subject to cotranslational mRNA decay. To inhibit the translation of multiple FZNF mRNAs, we utilized a previously reported MO that targeted a highly conserved 5′ UTR sequence upstream of the start codon of more than 200 C2H2-ZF genes on chromosome 4 (znfMO4) [66]. We also designed another MO complementary to the first 8 codons of 240 annotated FZNF ORFs (FiNZ ATGMO) (Fig 4C). Injection of either MO into wild-type fertilized eggs resulted in specific up-regulation of FZNF mRNAs at 6 hpf (Fig 4F). Changes in pre-mRNA levels were limited in the presence of specific MOs, consistent with the expected effect of MOs on translation inhibition. These results demonstrated that Znf598 posttranscriptionally reduces the levels of zygotic FZNF mRNAs during the MZT.

## Translation of C2H2-ZF domains causes ribosome stalling and collision

Our RNA-Seq analysis and follow-up experiments suggested that the translation of C2H2-ZF mRNAs caused ribosome stalling and collision, which induced NGD in a Znf598-dependent manner. To validate this model, we examined whether ribosomes indeed stalled while translating the *znf236* ORF using a tandem reporter assay. In this assay, an ORF fragment to be tested was sandwiched between the *Renilla* luciferase (Rluc) and firefly luciferase (Fluc) ORFs, each separated by the P2A ribosome skipping sequence (Fig 5A). Injection of this reporter mRNA into fertilized eggs enables the evaluation of the ribosome elongation rate at the inserted sequence as the ratio of luciferase activities [38]. We divided the entire ORF of *znf236* into 4

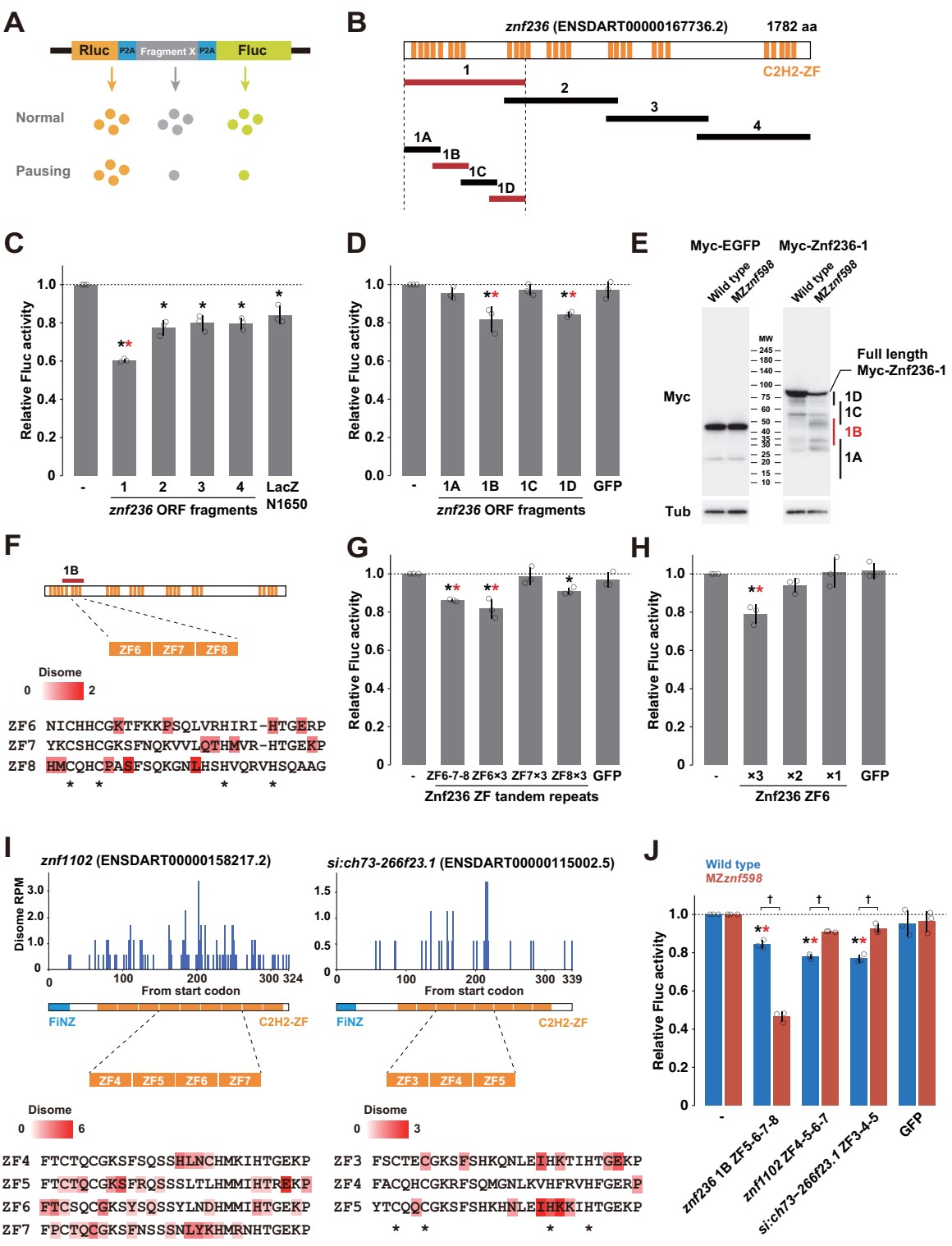

**Fig 5. Analysis of ribosome stalling sequences in C2H2-ZF mRNAs.** (A) A schematic of the tandem ORF assay. The Rluc ORF (orange) and Fluc ORF (light green) are separated by 2 P2A translation skipping sequences (blue). The sequence to be tested (shown as X in gray) is inserted between the 2 P2A sequences. (B) Schematic of the *znf236* ORF fragments analyzed in this study. C2H2-ZF is indicated in orange. Fragments that caused significant ribosome stalling in the tandem luciferase assay are indicated in red. (C, D) Results of the tandem ORF assay with the *znf236* fragment. (E) Western blotting to detect Myc-tagged EGFP (left panel) or Myc-tagged

Znf236 fragment 1 in the wild type and MZ*znf598* at 6 hpf. Tubulin (Tub) was detected as a loading control. The predicted positions of Myc-tagged peptides truncated in the Znf236 1A-1D regions are indicated on the right. (F) A detail of the 1B fragment. The amino acid sequences of ZF6, ZF7, and ZF8 are shown below. A-site positions of the stalled leading ribosome and corresponding footprint amounts are indicated as a red gradient. (G) Results of the tandem ORF assay with ZF6, ZF7, or ZF8 repeats. (H) Results of the tandem ORF assay with different numbers of ZF6 repeats. (I) Distributions of disome footprints (blue) in the *znf1102* and *si:ch73-266f23.1* ORFs. The amino acid sequences of C2H2-ZFs with high disome footprint amounts are shown below. A-site positions of the stalled leading ribosome and corresponding footprint amounts are indicated as a red gradient. (J) Results of the tandem ORF assay with 1B fragment (ZF5-6-7-8) of *znf236*, ZF4-5-6-7 of *znf1102*, and ZF3-4-5 of *si:ch73-266f23.1* in wild-type (blue) and MZ*znf598* (red) embryos. The graphs in C, D, G, H, and J represent the average of 3 independent experiments. Relative Fluc signals normalized to Rluc signals are shown. Values with no insert (-) were set to one. The error bars show the standard deviation. The open circles show each data point. Black asterisks indicate $p < 0.05$ compared to no insert. Red asterisks indicate $p < 0.05$ compared to LacZ-N1650 or GFP (Dunnett's test). Daggers indicate $p < 0.05$ (Student's *t* test). The numerical data underlying this figure can be found in S1 Data. hpf, hours postfertilization.

fragments of comparable length (fragments 1–4, approximately 1,400 to 1,600 nt) and subjected them to the tandem reporter assay. Compared to a control fragment with a similar length encoding the N-terminal region of LacZ (1,650 nt), fragment 1 significantly inhibited ribosome elongation compared to the other fragments (Fig 5B and 5C). Further analysis of fragment 1 revealed that 2 smaller fragments (fragments 1B and 1D, approximately 480 nt) inhibited elongation compared to GFP ORF (729 nt) (Fig 5B and 5D).

To detect ribosome stalling in the 1B region more directly, we injected an mRNA encoding N-terminally Myc-tagged *znf236* fragment 1 into wild-type and MZ*znf598* fertilized eggs and analyzed the Myc-tagged protein product by western blotting. While the expression of control Myc-EGFP was unchanged, the expression of the full-length Myc-Znf236 fragment 1 was reduced in the MZ*znf598* embryos (Fig 5E). In addition, we observed an accumulation of smaller products attributable to the peptides that were truncated within the 1B region. These results suggest that ribosomes frequently stall in the 1B region in a nonproductive state and that Znf598 rescues those stalled ribosomes.

Next, we assessed ribosome collision in the 1B region with previously published disome profiling data from the wild type at 4 hpf [41]. Since a subset of C2H2-ZF sequences shares highly homologous sequences at the nucleotide level [61,65], we utilized only uniquely mapped reads containing up to a single mismatch to the reference genome in this study (zebrafish genome assembly GRCz11). As a result, we detected fewer disome peaks in the new analysis than in the previous study (35% fewer peaks in all protein-coding mRNAs and 89% fewer peaks in C2H2-ZF mRNAs) (S4A Fig). We then sequenced 2 additional disome footprint libraries and merged the results with those of the previous 2 libraries to increase the footprint coverage (see Methods). Although disome footprint reads mapped to the *znf236* ORF was not abundant, 30.5% (18/59) of the reads were clustered in the 1B region (S4B Fig, left). These results suggest that ribosomes collide while translating the 1B region of the *znf236* ORF.

The 1B fragment encodes 4 C2H2-ZFs, and disome footprints were distributed around 3 C2H2-ZFs: ZF6, ZF7, and ZF8 (Fig 5F). Consistent with the disome profiling results, a fragment encoding 3 C2H2-ZFs (ZF6-7-8) was sufficient to induce ribosome stalling in the tandem reporter assay (Fig 5G). Ribosome stalling in the C2H2-ZF sequence occurred in a sequence-specific manner because 3 repeats of ZF6 induced ribosome stalling stronger than the repeats of ZF7 or ZF8 (Fig 5G). At least 3 copies of ZF6 were required to cause significant ribosome stalling, suggesting the importance of tandem C2H2-ZFs in causing ribosome stalling (Fig 5H).

Having validated ribosome stalling and collision in the C2H2-ZF sequences of *znf236*, we extended our investigation to the C2H2-ZF sequences of zygotic FZNF genes. We selected 2 FZNF genes validated by qRT-PCR (*znf1102* and *si:ch73-266f23.1*), in which disome footprints were distributed across the ORF at biased densities on C2H2-ZFs (Figs 5I and S4B). We then

isolated C2H2-ZF sequences with higher disome densities (ZF4-5-6-7 from *znf1102* and ZF3-4-5 from *si:ch73-266f23.1*) and compared them to the 1B fragment of *znf236* (ZF5-6-7-8) via the tandem luciferase reporter assay in wild-type and MZ*znf598* embryos. These C2H2-ZF sequences induced ribosome stalling at comparable levels to the 1B fragment of *znf236* in wild-type embryos (Fig 5J). In contrast, their effects differed in MZ*znf598* embryos. The 1B fragment of *znf236* caused a stronger stall in MZ*znf598* embryos, consistent with the western blotting analysis. On the contrary, 2 fragments from FZNF genes showed reduced stalling activity in MZ*znf598* embryos, likely representing a readthrough of the stall site in the absence of Znf598 reported in previous studies [22,24–26,67] (see Discussion). These results indicate that specific tandem C2H2-ZF sequences induce ribosome stalling and collision targeted by Znf598 in zebrafish embryos.

## Ribosome collision at C2H2-ZF induces mRNA decay

Following the detection of ribosome collision in model C2H2-ZF sequences, we next investigated the connection between ribosome collision and mRNA degradation. To this end, we performed a reporter mRNA injection assay that detected Znf598-dependent mRNA decay induced by the ribosome stall sequence of hCMV *gp48* uORF2 [38] (Fig 6A). We selected 1A-1D fragments of the *znf236* ORF as test cases and inserted them at the 3′ end of the sfGFP ORF. mRNA injection and qRT-PCR revealed that the insertion of fragment 1B destabilized the reporter mRNA in the wild-type embryos but not in the MZ*znf598* embryos (Fig 6B). Furthermore, 3 repeats of ZF6 (ZF6×3) were sufficient to recapitulate Znf598-dependent mRNA destabilization (Fig 6C). Injection of an MO blocking the translation initiation of the GFP ORF [38,50] into wild-type embryos stabilized the ZF6×3 reporter mRNA to a level comparable to that in MZ*znf598* embryos (Fig 6C), demonstrating that the decay was translation dependent, as it was the case for endogenous *znf236* mRNA (Fig 3E). These results showed that the specific C2H2-ZF sequence of the *znf236* mRNA induces both ribosome stalling and mRNA decay in zebrafish embryos.

To evaluate the connection between ribosome collision and mRNA decay more broadly, we analyzed the correlation between disome profiling data and RNA-Seq data. We initially focused on disome footprint peaks, as described in a previous study (S4A Fig) [41], and detected 65 disome peaks in 37 C2H2-ZF genes expressed at 6 hpf in wild-type embryos. These genes tended to be more up-regulated than the remaining C2H2-ZF genes in the MZ*znf598* embryos (S5A Fig). On the other hand, we failed to detect disome peaks in many disome dense regions in the C2H2-ZF sequences (e.g., *znf236*, *znf1102*, and *si:ch73-266f23.1*). This discrepancy occurred because the disome peak analysis aimed to detect strong and consistent ribosome collision at a single codon rather than the high frequency of ribosome collisions in broader regions. Considering these points, we devised a disome localization score, which is the relative accumulation of disome footprints on C2H2-ZF sequences of a given ORF normalized by the total disome footprints formed on the entire ORF (Fig 6D). This domain-wise evaluation gave higher scores if disome footprints were clustered on C2H2-ZF sequences, whereas stochastic or C2H2-ZF-independent disome formation was reflected by lower scores. Based on the disome localization score, we divided the C2H2-ZF mRNAs with disome footprints into 3 groups (high, middle, and low) (S5B Fig). There was no bias in the disome localization score according to the footprint read amount (S5C Fig). We found that C2H2-ZF mRNAs in the high group were more up-regulated in the MZ*znf598* embryos than those in the middle and low groups (Fig 6E). This genome-wide correlation supports the model wherein ribosome collision in C2H2-ZF sequences induces mRNA decay in zebrafish embryos (Fig 6F).

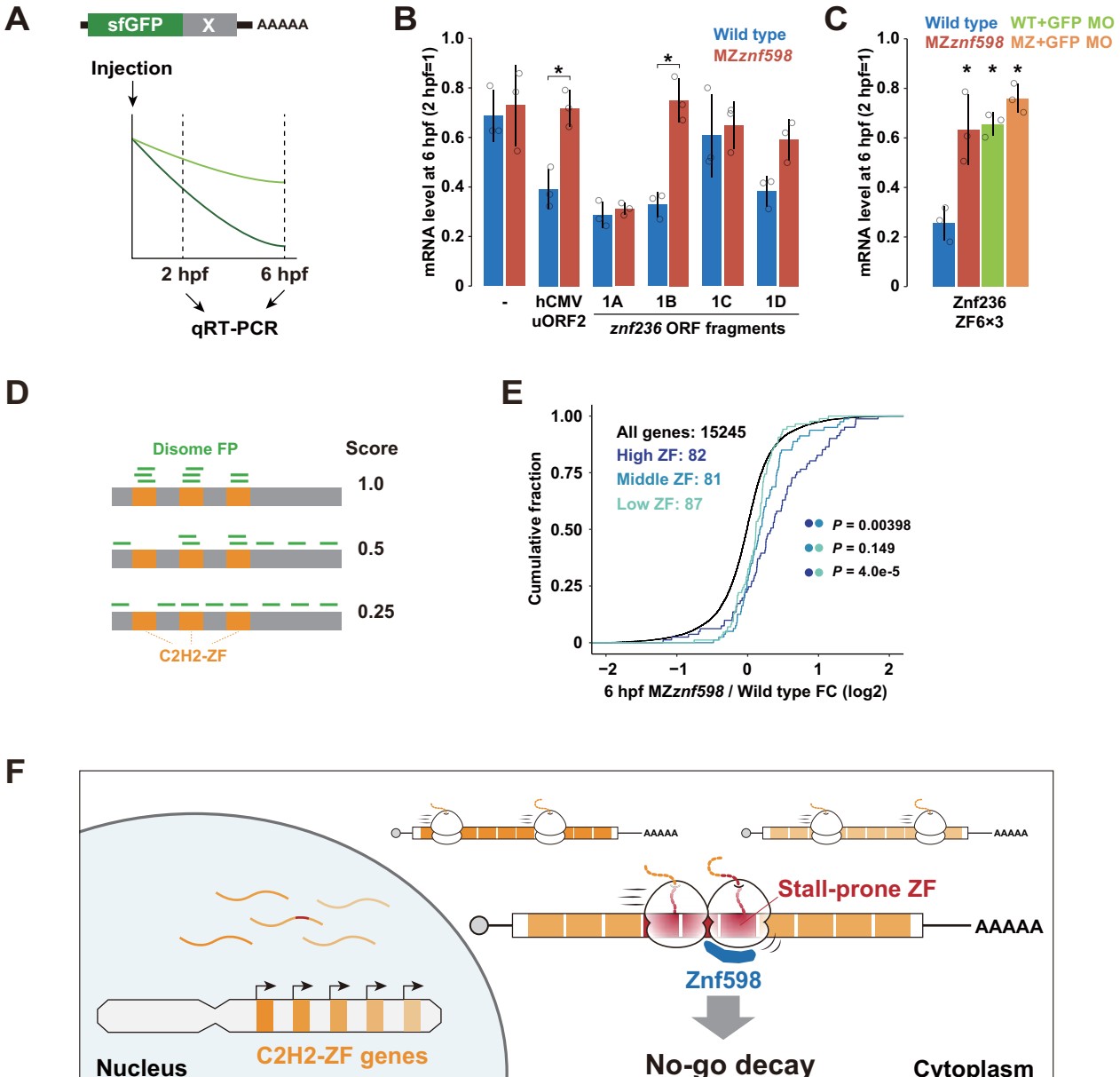

**Fig 6. Analysis of the effect of C2H2-ZF sequences on mRNA stability.** (A) A scheme of the mRNA injection assay used to measure the effect of the sequence to be tested (shown as X in gray) on mRNA stability. (B) qRT-PCR analysis of injected sfGFP reporter mRNAs at 6 hpf relative to 2 hpf in wild-type (blue) and MZ*znf598* (red) embryos. (C) qRT-PCR analysis of injected sfGFP-ZF6×3 reporter mRNAs at 6 hpf relative to 2 hpf in wild-type (blue), MZ*znf598* (red), GFP MO-injected wild-type (light green), and GFP MO-injected MZ*znf598* (orange) embryos. The graphs in B and C represent the average of 3 independent experiments. The error bars show the standard deviation. The open circles show each data point. Asterisks indicate $p < 0.05$ (Student's *t* test in B and Dunnett's test compared to the wild type in C). (D) A scheme of the disome localization score analysis. (E) Cumulative distributions of fold changes in mRNA levels in MZ*znf598* embryos compared to wild-type embryos at 6 hpf. All genes (black) and C2H2-ZF genes with high (dark blue), middle (blue), and low (turquoise) disome localization scores are shown. The x-axis shows the fold change, and the y-axis shows the cumulative fraction. The *p* values are shown on the right (Kolmogorov–Smirnov test). (F) A model of NGD for degrading mRNAs encoding stall-prone tandem C2H2-ZF sequences in zebrafish. The numerical data underlying this figure can be found in S1 Data. hpf, hours postfertilization; MO, morpholino oligonucleotide; NGD, no-go decay.

## Sequence features that induce ribosome stalling at C2H2-ZF

To understand the sequence features of C2H2-ZF that cause ribosome collision, we determined the position of the leading ribosome in disome footprints relative to the C2H2-ZF sequence. Approximately 85% of zebrafish C2H2-ZFs are 28 amino acids in length (8168/9622 C2H2-ZFs) and contain cysteine and histidine, which accommodate a zinc ion, at invariant positions (S5D Fig) [60,68]. Disomes were formed at various positions in the C2H2-ZF sequence and did not converge at a specific position (S5E Fig). Consistent with these results, disome occupancies at the codons on which A-, P-, and E-sites in the leading ribosome lay were not drastically skewed in C2H2-ZF sequences compared to entire ORF sequences (S5F Fig). Therefore, ribosomes are unlikely to stall and collide at a consistent position or codon shared in C2H2-ZF sequences.

Electrostatic interactions between the negatively charged ribosome exit tunnel and a positively charged nascent polypeptide are known to cause ribosome stalling and collision [41,42,44,69–71]. Notably, C2H2-ZF proteins presented high isoelectric points (pIs), and C2H2-ZF genes up-regulated in the MZ*znf598* embryos encoded proteins with even greater pIs (S5G Fig). Consistent with these data, amino acid sequences encoded upstream of the leading ribosome that stalled at C2H2-ZF had a greater mean net charge than did the control sequences randomly sampled from C2H2-ZF (S5H Fig, between 14 and 29 amino acids upstream of the A site). These data indicate that ribosomes stalled in C2H2-ZF sequences tend to hold positively charged peptides in their exit tunnel, possibly contributing to ribosome stalling.

In addition to nascent peptides held in ribosomes, codon sequences also affect the ribosome traversal [72]. Interestingly, genome-wide prediction of mRNA stability by iCodon, which accounts for the effect of codons on mRNA stability, revealed that mRNAs encoding zinc finger transcription factors are significantly unstable in humans [73]. iCodon predicted that zebrafish C2H2-ZF genes were significantly unstable, and C2H2-ZF genes up-regulated in MZ*znf598* were predicted to be even less stable (S5I Fig). Since the effect of codons on mRNA stability is strongly related to the codon decoding rate by the ribosome [38,51,74–76], this prediction indicates that C2H2-ZFs are enriched with codons disadvantageous for translation elongation. Hence, 2 features that increase the risk of ribosome stalling, positively charged peptides and nonoptimal codons, are inherent in C2H2-ZF sequences.

## Evaluation of ribosome stalling at tandem C2H2-ZF sequences in human cells

Finally, we addressed whether the stall-prone nature of tandem C2H2-ZF and degradation of C2H2-ZF mRNAs by ZNF598 is conserved in humans. To this end, we analyzed mRNA half-lives using BRIC-Seq in HEK293T cells expressing control shRNA or ZNF598-targeting shRNAs (S6A Fig) [41,77]. Among the 3,070 transcripts with reliable half-lives, 169 transcripts encoded C2H2-ZF, and 65 encoded RING-ZF. We found that the half-life of transcripts encoding C2H2-ZF tended to be elongated in ZNF598 knockdown cells, whereas that of RING-ZF did not (S6B Fig). We then analyzed the published disome profiling data from HEK293 cells [41]. Disomes formed at C2H2-ZF sequences had no clear positional preference in HEK293 cells, as observed in zebrafish (S6C and S6D Fig). The mean net charge of nascent polypeptides translated by leading ribosomes stalled at human C2H2-ZF sequences showed a high net charge; however, the high net charge region was more proximal to the decoding/peptidyl transferase centers in humans (S6E Fig).

We selected 3 mRNAs containing disome-rich regions overlapping with tandem C2H2-ZF domains (ZNF768, ZBTB7A, and ZNF100) and performed validation experiments (S7A–S7C

Fig). Two out of 4 tandem C2H2-ZF fragments caused detectable ribosome stalling in the dual luciferase assay in HEK293T cells (S7D Fig). Nevertheless, time course qRT–PCR analysis after transcription inhibition by Actinomycin D showed similar mRNA decay patterns in control and ZNF598 KD cells (S7E and S7F Fig). Therefore, while the stall-prone nature of C2H2-ZF is common in zebrafish and humans, the contribution of ribosome stalling at C2H2-ZF to the mRNA stability in human cells requires further investigation.

## Discussion

Toward a comprehensive understanding of ribosome collision and its quality control landscape, we identified mRNAs encoding C2H2-ZF as targets of Znf598-dependent mRNA decay in zebrafish embryos. Since C2H2-ZF sequences are prone to ribosome stalling and disome formation, our study suggested that C2H2-ZF sequences are endogenous ribosome stallers that induce NGD in zebrafish embryos (Fig 6F). Information on NGD and its substrate mRNA is still limited in vertebrates compared to yeast and worms. In future studies, C2H2-ZF mRNA will be a good model for characterizing NGD in vertebrates.

Disome profiling in yeast and mammalian cells revealed fundamental sequences of codons and amino acids associated with ribosome collision sites [41–44]. Such stall-prone sequences are often observed between or outside the protein domains [41,44], likely representing programmed ribosome slowdown assisting cotranslational protein folding [78]. In contrast, our understanding of broader protein domains or motifs that are prone to ribosome collision is limited, except for the signal sequences [44,79]. Therefore, it is intriguing that C2H2-ZF, the most abundant DNA-binding domain in the eukaryotic genome [60,80], was identified as a prevalent collision-inducing sequence. The broad distribution of disomes in the C2H2-ZF sequence likely hindered its identification as a typical collision site in previous disome analyses [41–44], which have focused mainly on strong and consistent collision sites [41–44]. We speculate that revisiting disome profiling data with the disome density analysis in broader regions will reveal unidentified ribosome stallers. In addition, our study highlighted the importance of combining disome profiling with the analysis of functional consequences, e.g., NGD and RQC, for understanding ribosome collision events that were not characterized by disome profiling alone.

The identification of C2H2-ZF mRNAs as endogenous NGD targets raises questions about the underlying mechanism of ribosome stalling. Since stall-prone C2H2-ZF sequences lack well-established ribosome stalling motifs (e.g., a stretch of lysine AAA codons), they might represent a mechanism distinct from known examples of ribosome collision (S5D and S6C Figs) [10,13,37,71,81,82]. As clues to understanding the stalling mechanism, we showed that basic amino acids and nonoptimal codons were enriched in C2H2-ZF sequences prone to NGD (S5G–S5I Fig). A plausible scenario is that the tandem arrangement of the positively charged C2H2-ZF sequence causes sequential exposure of the positively charged nascent polypeptides to the ribosome exit tunnel and synergistically hampers ribosome traverse. The presence of nonoptimal codons in such a sequence context could further increase the risk of ribosome collision. Whether this model explains the general mechanism of ribosome stalling in diverse C2H2-ZF sequences awaits further validation. Other features, including inhibitory codon and amino acid pairs, cotranslational folding, and interactions with chaperones, might be associated with tandem C2H2-ZF sequences and synergistically contribute to ribosome stalling [83–86]. Elucidation of the detailed ribosome stalling mechanism at the tandem C2H2-ZF sequence will expand our knowledge of ribosome stalling mechanisms in eukaryotes.

Although the stall-prone nature of widespread C2H2-ZF sequences has been revealed, its biological impact has not been fully elucidated. MZ*znf598* embryos showed an elevated level of

phosphorylated eIF2α but developed without apparent morphological defects, suggesting that the failure of RQC or NGD at C2H2-ZF sequences does not perturb the cellular translation status to a catastrophic level (Figs 1 and S1). This result is consistent with the normal growth phenotype of the yeast Hel2 mutant [87] and presumably due to the presence of multiple mechanisms that act redundantly to solve or mitigate translation elongation defects, such as GCN1/2, ZAKα, and GTPBP1/2 [56,88–91]. At the individual transcript level, persistent ribosome collision in the absence of Hel2/ZNF598 could either increase the expression of the full-length protein by allowing readthrough of the stall site or decrease the protein output by road-blocking the following ribosomes [22,24–26,67,92]. Indeed, we observed the latter scenario with a C2H2-ZF sequence of Znf236 (Fig 5E and 5J), and the former scenario with 2 FZNF C2H2-ZF sequences (Fig 5J). These data support the idea that the consequence of the loss of RQC/NGD could be context dependent [14]. mRNA stabilization due to the lack of NGD and loss of the initiation block by the GIGYF-4EHP pathway additionally affect protein output [35,93,94]. As such, the change in the expression of each stall-prone C2H2-ZF protein in the absence of Znf598 is not predictable. In addition, the rapid evolution of C2H2-ZF genes has made it difficult to understand the impact of ribosome collision on protein function. Notably, a recent study proposed the role of FZNF proteins in transposon silencing during MZT [65]. Therefore, perturbation of FZNF expression by the loss of Znf598 might affect transposon silencing and cause genetic alterations on a longer timescale. Alternatively, NGD might contribute to generating the sharp expression peak of FZNFs at ZGA by promoting the clearance of their mRNAs [63,65]. In support of the importance of restricting C2H2-ZF expression, aberrant expression of C2H2-ZF genes has been observed in the zebrafish *kdm2aa* mutant that develops melanoma [95]. Understanding the impact of ribosome collision events in each C2H2-ZF sequence on protein output and phenotypic outcome in embryonic and postembryonic stages is an important topic for future studies.

The high propensity of the widely shared C2H2-ZF sequence to stall the ribosome is a paradoxical phenomenon. The C2H2-ZF domain is the largest protein domain family in the eukaryotic genome, most of which are present as tandem arrays in a protein [60,68]. The evolution of tandem C2H2-ZF genes is highly dynamic, and species-specific amplification of the C2H2-ZF gene is frequently observed [60,61,96]. Accordingly, the repertoire of C2H2-ZF has expanded rapidly, and each species has vast diversity in C2H2-ZF genes [60,97]. On the one hand, the rapid evolution and diversification of the C2H2-ZF sequences are beneficial for expanding the DNA-binding capacity of the proteome to regulate endogenous genes and transposable elements [60,65,97,98]. On the other hand, it could drive the emergence of stall-prone C2H2-ZF sequences at a rate exceeding evolutionary pressure. In this regard, one possible role of NGD is maintaining the quality of the translatome during evolution by reducing the expression of transiently and frequently acquired stall-prone C2H2-ZF sequences. A comprehensive multispecies analysis is necessary to understand the role of RQC and NGD in the regulation of C2H2-ZF genes during protein evolution.

Finally, our study implies a broad impact of NGD on the expression of both endogenous and artificial repetitive sequences. C2H2-ZF has been engineered as a platform for programmable DNA-binding modules, such as zinc finger nucleases [99]. Elucidation of the ribosome stall mechanism involving the C2H2-ZF sequence would help in the design of efficient DNA-binding modules by preventing stall-prone C2H2-ZF. We also speculate that other tandem repeat domains, which are abundant in the proteome and diversify rapidly [100], harbor a risk of acquiring ribosome stall sequences. Further studies will elucidate the role of NGD and RQC in the evolution and regulation of mRNAs encoding stall-prone tandem repeats.

## Materials and methods

### Zebrafish

The zebrafish AB strain was used as the wild type. The zebrafish *znf598* mutant strain harboring an 11 bp deletion in exon 1 was described previously [38]. The zebrafish *rps10* K138/139R mutant strain was described in [55]. Fish were raised and maintained at 28.5˚C under standard laboratory conditions according to the Animal Experiment Protocol (2023–37) approved by The Kyoto Sangyo University Institutional Animal Care and Use Committee. Fertilized eggs were obtained by natural breeding. Embryos were developed in system water at 28.5˚C. Larvae were fed from 5 dpf after hatching. To compare the viability and growth of wild-type and MZ*znf598* fish, 20 MZ*znf598* embryos were mixed with 20 AB embryos at 1 dpf and raised together in a single tank as described previously [101]. The body length of individual fish was measured and photographed at the indicated times, and the genotype was determined by fin-clipping and genotyping PCR using the primers listed in S2 Table. Bright-field images were acquired using a SteREO Lumar V12 microscope and an AxioCam MRc camera with ZEN software (Zeiss, Jena, Germany) in spline mode.

### Microinjection

Myc-tagged Znf598 mRNAs and tandem ORF luciferase mRNAs were transcribed from a linearized plasmid DNA template using the mMESSAGE mMACHINE SP6 Transcription Kit (Thermo Fisher Scientific) and purified using the RNeasy Mini Kit (QIAGEN). sfGFP, Myc-tagged EGFP, and Myc-tagged Znf236 mRNAs were transcribed from a linearized plasmid DNA template using the SP6-Scribe Standard RNA IVT Kit (CELLSCRIPT), the m$^7$G cap structure was added to the purified RNA using the ScriptCap m$^7$G Capping System (CELLSCRIPT), the poly(A) tail was added using the A-Plus Poly(A) Polymerase Tailing Kit (CELLSCRIPT), and the mRNA was purified using the RNeasy Mini Kit. Purified mRNAs were diluted in water to the following concentrations: tandem luciferase mRNAs and sfGFP reporter mRNAs: 50 ng/μl and Myc-Znf598 mRNAs: 200 ng/μl. For injection of Myc-tagged EGFP or Znf236 mRNAs, 50 ng/μl of mRNA and 60 ng/μl QuantiLum Recombinant Luciferase protein (Promega) were coinjected. Microinjection was performed as described previously [38,50] using an IM300 Microinjector (NARISHIGE). Approximately 1,000 pl of the solution was injected per embryo within 15 min of fertilization.

### Plasmids

Plasmids for preparing dual-luciferase mRNAs were constructed based on the pLuc plasmid [38]. DNA fragments encoding test sequences were prepared by PCR, annealing complementary oligonucleotides, or DNA fragment synthesis (IDT) and cloned between the EcoRI and XhoI sites of pLuc. The PCR primers used are listed in S2 Table. To design DNA fragments encoding artificial zinc finger repeats (ZF6×2, 6×3, 7×3, and 8×3), synonymous codon substitutions were introduced to avoid repeat sequences that prohibited DNA fragment synthesis. To construct plasmids for preparing sfGFP mRNAs, DNA fragments encoding test sequences were cloned between the EcoRI and EcoRV sites of the pCS2+neo-sfGFP-*suv39h1a* 3′ UTR [38]. To construct plasmids encoding Myc-tagged EGFP and Znf236 fragment 1, ORFs were amplified via PCR and subsequently inserted into pCS2+MT between the NcoI and EcoRI (EGFP) or EcoRI and XhoI (Znf236 fragment 1) sites. The pCS2+MT-Znf598 plasmids were described previously [38].

### RNA-Seq

RNA-Seq was performed as described previously [38]. Briefly, total RNA was extracted from 40 to 50 wild-type and MZ*znf598* zebrafish embryos at 1 and 6 hpf using TRI Reagent

(Molecular Research Center). After DNaseI treatment and purification by the RNeasy Mini Kit, RNA integrity was confirmed using an Agilent RNA 6000 Nano Chip (Agilent Technologies, United States of America). Total RNA samples were treated with a Ribo-Zero Gold rRNA Removal Kit (Human/Mouse/Rat) (Illumina, USA) and used for RNA-Seq library preparation using an Illumina TruSeq Stranded Total RNA Library Prep Kit (Illumina) with multiplexing. The pooled libraries were sequenced on a NextSeq500 (Illumina) platform with 76 bp single-end sequencing.

## Ribosome footprint profiling

A portion of the zebrafish monosome and disome footprint sequencing data at 4 hpf used in this study were published previously [41] (GSE133392). We obtained 2 additional disome footprint libraries at 4 hpf, corresponding to the monosome footprint libraries [38] (GSE173604), to increase the depth of disome footprint reads. These additional libraries were prepared in parallel to the published libraries following the protocol described in [102].

## Analysis of RNA-Seq data

Zebrafish RNA-Seq data were analyzed using Galaxy on public servers [103]. Single-ended reads were preprocessed by fastp [104] (version 0.23.2+galaxy0) and mapped to the zebrafish genome reference GRCz11(danRer11) using STAR [105] (version 2.7.8a+galaxy1) by allowing up to a single-nucleotide mismatch with the—outFilterMultimapNmax 1 –outFilterMismatchNmax 1 options. Only uniquely mapped reads were subjected to downstream analysis. Mapped reads were counted by featureCounts [106] (version 2.0.1+galaxy2). To analyze the reads mapped to introns, featureCounts was performed with the GFF feature type filters "transcript" and "exon" with the -O option, and the latter was subtracted from the former. Differential expression was analyzed by edgeR [107] (version 3.36.0+galaxy0). Only genes with Ensemble Canonical transcript annotation were analyzed as protein-coding genes in this study. Genes with at least 1 cpm were considered expressed at detectable levels and were used for further analysis. For the identification of differentially expressed genes, the FDR was set to 0.01. Functional enrichment analysis of the differentially expressed genes was performed using DAVID [108], and mRNAs degraded via maternal and zygotic decay pathways were classified based on a previous study [50]. Maternal mRNAs with high levels of $m^6A$ modification were classified following previous studies [53,54].

## Protein domain search and pI analysis

Protein domains in zebrafish and human amino acid sequences were detected using pfsearch [109] (version 2.3.5) based on matrix profiles obtained from the PROSITE database (https://prosite.expasy.org) [110]. The entry IDs of the matrix profiles used are as follows: PS50157 for the C2H2-ZF domain and PS50089 for the RING-ZF domain. The pI of the zebrafish proteins was calculated using the Peptides package in R.

## mRNA stability prediction by iCodon

The mRNA stability of the zebrafish genes was predicted from the protein-coding sequences using iCodon algorithm [73]. Only sequences with a length multiple of 3 were used for the analysis.

## Analysis of ribosome profiling data

Ribosome profiling data were processed as previously described [41,111,112] with minor changes. Briefly, reads were preprocessed using fastp [104] and mapped to the zebrafish

(GRCz11) or human (GRCh38/hg38) reference genome using STAR [105] by allowing up to a single-nucleotide mismatch and allowing mapping to a single locus with the—outFilterMulti-mapNmax 1 –outFilterMismatchNmax 1 options. Disome occupancy was calculated as the ratio of reads at given codons to the average reads per codon on all ORFs or on C2H2-ZF domains. The disome localization score was calculated as the ratio of reads on C2H2-ZF domains in an ORF to the total number of reads on the entire ORF. Transcripts with at least 10 reads on the ORF were subjected to downstream analysis. Since most C2H2-ZF domains are 28 amino acids in length, disome occupancies and disome localization scores were calculated for C2H2-ZF domains 28 amino acids in length.

The net charge of a nascent polypeptide associated with the stalled leading ribosome was calculated with 6-amino-acid sliding windows using the Peptides package in R. A-site positions of the leading ribosomes in 28-amino-acid C2H2-ZF domains were used for the net charge calculation (experiment). For the control, the same number of A-site positions in 28-amino-acid C2H2-ZF domains was randomly chosen, and their net charge was calculated (simulation).

## BRIC-Seq

**Library preparation.** BRIC-Seq was performed as described previously with modifications [77]. The HEK293T cells expressing control or ZNF598 shRNA [41] were seeded in 10 cm dishes and incubated in 10 ml of DMEM supplemented with 150 μM bromouridine (BrU, FUJIFILM Wako Pure Chemical Corporation) for 24 h. The cells were then washed twice with 5 ml of DMEM without BrU, incubated in 10 ml of DMEM without BrU for 0, 2, 4, 6, 10, or 24 h, washed with 5 ml of PBS, and lysed by adding 600 μl of TRIzol reagent (Thermo Fisher Scientific) directly to the dishes. Total RNA was purified using a Direct-zol RNA MiniPrep Plus Kit (Zymo Research).

BrU-containing spike-in mRNAs were synthesized via in vitro transcription. The template DNA fragments were PCR-amplified using the psiCHECK2 plasmid (Promega) with 5′-TAA TACGACTCACTATAGG-3′ and 5′-CACACAAAAAACCAACACACAG-3′ (Rluc); the psi-CHECK2 plasmid with 5′-ACTTAATACGACTCACTATAGGAAGCTTGGCATTCCGG-3′ and 5′-TGTATCTTATCATGTCTGCTCGAAG-3′ (Fluc); and the pColdI-GFP plasmid [113] with 5′-TGACTAATACGACTCACTATAGGATCTGTAAAGCACGCCATATCG-3′ and 5′-TGGCAGGGATCTTAGATTCTG-3′ (GFP). In vitro transcription was performed using a T7-Scribe Standard RNA IVT Kit (CELLSCRIPT) in the presence of 1.2 mM BrUTP (Jena Bioscience) and 5 mM UTP. The RNA was purified with RNAClean XP beads (Beckman Coulter).

For immunoprecipitation, 25 μl of Dynabeads M-280 Sheep Anti-Mouse IgG (Thermo Fisher Scientific) was incubated with 1.5 μg of anti-BrdU antibody (BD Biosciences, 555627) for 3 h at 4˚C and equilibrated with BRIC-Seq lysis buffer (lysis buffer without cycloheximide and chloramphenicol). Twenty micrograms of total RNA mixed with a spike-in RNA mixture (1 ng of Rluc, 0.2 ng of Fluc, and 0.04 ng of GFP) was incubated with the magnetic beads in BRIC-Seq lysis buffer for 2 h at 4˚C, washed 5 times with 100 μl of BRIC-Seq lysis buffer, and mixed with 200 μl of TRIzol reagent. RNA was extracted with a Direct-zol RNA MicroPrep Kit and subjected to an RNA-Seq library preparation using a Ribo-Zero Gold rRNA Removal Kit (Human/Mouse/Rat) (Illumina) followed by a TruSeq Stranded Total RNA Kit (Illumina). Libraries were sequenced on a HiSeq 4000 (Illumina) with the single-end/50 nt-long read option.

**Data analysis.** Filtering by quality and adapter trimming were performed using fastp [104]. Reads were subsequently aligned to the noncoding RNAs using STAR 2.7.0a [105] and

excluded from analysis. The remaining reads were mapped to the human genome hg38 with the annotation of the GENCODE Human release 32 reference using STAR 2.7.0a with—out-FilterMismatchNmax 1—outFilterMultimapNmax 1 options. The read counts for each transcript were analyzed by featureCounts [106].

The fold changes in read counts compared to those of the 0 h samples were calculated with the DESeq2 package [114] and then renormalized to the mean change in 3 spike-ins. For the calculation of the mRNA half-life, the $\log_2$-converted fold-change at each time point was fitted to a linear model. Since the decay of transcripts sometimes slows at later time points, we calculated the half-life using all combinations of earlier samples (i.e., 0–4 h, 0–6 h, 0–10 h, and 0–24 h) and fitted the results with the maximum $R^2$ value. Transcripts with $R^2 > 0.9$ and $0 <$ half-life $< 24$ h were used in downstream analyses.

## qRT-PCR

qRT-PCR analysis of zebrafish mRNAs was performed as previously described [38]. Briefly, total RNA was prepared using TRI Reagent (Molecular Research Center) and cDNA was synthesized using the PrimeScript RT reagent kit with gDNA eraser (TaKaRa). A random hexamer was used for cDNA synthesis. qRT-PCR was performed using SYBR Premix Ex TaqII (Tli RNaseH Plus) and the Thermal Cycler Dice Real Time System (TaKaRa). The data were normalized using 18S rRNA as a reference.

For qRT-PCR analysis of human mRNAs, HEK293T cells in a 6-well plate were treated with 5 μg/ml Actinomycin D (FUJIFILM Wako Pure Chemical Corporation) for 0.5, 1, and 2 h. The cells were then washed with 1 ml of PBS and lysed in 100 μl of lysis buffer [20 mM Tris-HCl (pH 7.5), 150 mM NaCl, 5 mM MgCl2, 1 mM dithiothreitol and 1% Triton X-100]. Lysates were cleared by centrifugation at 20,000 g for 10 min at 4°C, and RNAs were extracted by TRIzol LS reagent (Thermo Fisher Scientific) and a Direct-zol RNA Microprep Kit (Zymo Research). After DNA digestion with 2 U of Turbo DNase (Thermo Fisher Scientific), the qRT-PCR was performed using ReverTra Ace qPCR Master Mix (TOYOBO) and a CFX Connect system (Bio-Rad) with iTaq Universal SYBR Green Supermix (Bio-Rad).

The sequences of primers used are listed in S2 Table.

## qPCR from genomic DNA

Genomic DNA was extracted from a single wild-type embryo at 24 hpf. qPCR analysis of genomic DNA was performed essentially as described for qRT-PCR, except for that genomic DNA was used as a template. Copy numbers of FiNZ exons and *rpl24* were estimated based on a standard curve of a serial dilution of a plasmid containing the amplicon. The copy number of the FiNZ exon was normalized to that of *rpl24* to calculate the copy number per haploid genome. The sequences of primers used are listed in S2 Table.

## PAT assay

The PAT assay was performed as previously described [50]. Briefly, 150 ng of total RNA was incubated with 75 U of yeast poly(A) polymerase (PAP) (Affymetrix) in the presence of GTP/ITP mixture at 37°C for 60 min. cDNA was synthesized at 44°C for 15 min using the Prime-Script RT reagent kit with gDNA eraser (TaKaRa) and the y300 PAT universal C10 primer. PAT-PCR was performed using a 3′ UTR-specific forward primer and a PAT universal primer with GoTaq Green Master Mix (Promega). The PCR products were separated by 6% PAGE in 0.5× TBE. The gels were stained with GelRed (Biotium), and the signals were detected using an Amersham Imager 680 (Cytiva).

## Luciferase assay

The luciferase assay in zebrafish embryos was performed as previously described [38]. Briefly, 5 to 10 embryos were collected at 3 hpf and lysed in Passive Lysis Buffer (Promega) containing cOmplete Protease Inhibitor Cocktail (Sigma–Aldrich). Ten microliters of lysate containing 1 embryo was used for the assay. The luciferase activities were measured using the Dual-Glo Luciferase Assay System and a GloMax 20/20 luminometer (Promega).

The luciferase assay in HEK293T cells was performed as previously described [115]. Briefly, cells were transfected with the plasmids using TransIT-293 reagent (Mirus), incubated for 24 h, and lysed in Passive Lysis Buffer. The signals were detected with the Dual-Luciferase Reporter Assay System (Promega) in GloMax Navigator (Promega).

The intensity of Fluc activity was normalized to the intensity of Rluc activity. The normalized Fluc activity was further normalized to that of the control sample (no insert).

## Western blotting

Zebrafish proteins were detected as described previously [38] using anti-Myc (MBL Life Science My3 mouse monoclonal, 1:4,000), anti-phosphorylated eIF2α (Ser51) (Cell Signaling Technology #3398 rabbit monoclonal, 1:1,000), anti-eIF2α (Cell Signaling Technology #9722 rabbit polyclonal, 1:4,000), and anti-α-tubulin-pAb HRP-DirecT (MBL Life Science PM054-7 rabbit polyclonal 1: 10,000) antibodies. To detect Myc-*znf236*, embryos were collected at 6 hpf, and an aliquot of the lysate was subjected to a luciferase assay to ensure the accuracy of the injection amount. Another aliquot of the same lysate was used for western blotting. The signals were developed using Luminata Forte (Millipore) and detected using an Amersham Imager 680 (Cytiva).

Human proteins were detected using anti-ZNF598 (Novus Biologicals #NBP1-84658 rabbit polyclonal, 1:1,000), anti-β-actin (Medical and Biological Laboratories #M177-3 mouse monoclonal, 1:1,000), IRDye 800CW anti-rabbit IgG (LI-COR #926–32211, 1:10,000), and IRDye 680RD anti-mouse IgG (LI-COR #925–68070, 1:10,000). The images were acquired with an ODYSSEY CLx imager (LI-COR).

## Supporting information

**S1 Fig. Additional analysis of the zebrafish *znf598* mutant.** (A) The genotyping results of 10-week-old siblings obtained by crossing heterozygous *znf598* mutant fish. The *p* value was calculated by the chi-square test. (B) Distributions of body length in 10-week-old wild-type, *znf598* heterozygous, and *znf598* homozygous sibling fish obtained by crossing heterozygous *znf598* mutant fish. The *p* values were calculated by one-way ANOVA test. (C) A scheme of the growth and survival test comparing wild-type and MZ*znf598* fish. (D, E) Distributions of the body length in wild-type and MZ*znf598* fish at 12, 16, and 28 weeks after birth. In (D), results of the 2 experiments were combined and plotted. In (E), results of a single experiment were plotted. The *p* values were calculated by the Mann–Whitney U test (two-tailed). The data underlying this figure can be found in S1 Data.
(TIF)

**S2 Fig. Additional information about maternal mRNA analysis in zebrafish embryos.** (A) A Venn diagram of NGD target candidates (yellow), maternal decay mRNAs (pink), and zygotic decay mRNAs (light blue). The numbers of genes in each category are shown. (B–D) Cumulative distributions of maternal decay mRNAs, zygotic decay mRNAs, and maternal mRNAs with a high level of m$^6$A modification. The x-axis shows the fold change in mRNA expression at 6 hpf compared to 1 hpf in wild-type and MZ*znf598* embryos, and the y-axis

shows the cumulative fraction. The *p* values calculated by the Kolmogorov–Smirnov test are indicated. (E) qRT-PCR analysis of *znf236* mRNA in wild-type (blue) and MZ*rps10*KR embryos (red) at 6 hpf relative to 2 hpf. *gstm* mRNA (miR-430 target) and *ddx4* mRNA (stable maternal mRNA) are shown as controls. The asterisk indicates $p < 0.05$ (Student's *t* test). The data underlying this figure can be found in S1 Data.
(TIF)

**S3 Fig. Additional information about the expression patterns of C2H2- ZF mRNAs in zebrafish.** (A) Chromosomal distributions of C2H2-ZF genes and their relative mRNA expression at 6 hpf compared to 1 hpf in wild-type embryos. The x-axis shows chromosome numbers, and the y-axis shows the fold change in mRNA levels at 6 hpf compared to 1 hpf. (B) Chromosomal distributions of C2H2-ZF genes and their relative mRNA expression in MZznf598 embryos compared to wild-type embryos at 6 hpf. The x-axis shows chromosome numbers, and the y-axis shows fold changes in mRNA levels. In A and B, the value of each C2H2-ZF gene is plotted as a dot. The box represents the interquartile range (IQR), with the median indicated by the thick horizontal line in the box. The whiskers represent the variation within 1.5 IQR outside the upper and lower quartiles. (C) Cumulative distributions of fold changes in intron RNA levels in MZznf598 embryos compared to wild-type embryos at 6 hpf. (D) Cumulative distributions of the mature mRNA fold change subtracted by the intron RNA fold change in MZznf598 embryos compared to wild-type embryos at 6 hpf. In C and D, C2H2-ZF genes on chromosome 4 (orange), C2H2-ZF genes on other chromosomes (purple), and genes without C2H2-ZF (black) are shown. The x-axis shows the fold change, and the y-axis shows the cumulative fraction. The p values are shown on the left (Kolmogorov–Smirnov test). The data underlying this figure can be found in S1 Data.
(TIF)

**S4 Fig. Additional information about disome profiling.** (A) Venn diagrams of disome peaks detected in a previous study [41] and in this study (1 mismatch and unique map). The number of peaks detected in each study is shown. Left: disome peaks in all genes. Right: disome peaks in C2H2-ZF genes. (B) Distributions of monosome (upper, gray) and disome (lower, blue) footprints on the *znf236*, *znf1102*, and *si:ch73-266f23.1* ORFs. The data underlying this figure can be found in S1 Data.
(TIF)

**S5 Fig. Additional information about disomes formed on zebrafish C2H2-ZF genes.** (A) Cumulative distributions of fold changes in mRNA levels in MZ*znf598* embryos compared to wild-type embryos at 6 hpf. All genes (black) and C2H2-ZF genes with disome peaks (green) or without disome peaks (light green) are shown. The x-axis shows the fold change, and the y-axis shows the cumulative fraction. The *p* values are shown on the left (Kolmogorov–Smirnov test). (B) A histogram showing the distributions of disome localization scores on C2H2-ZF genes. The x-axis shows the disome localization score, and the y-axis shows the gene count. (C) A scatter plot showing the disome localization score (x-axis) and total read number for each ORF (y-axis). (D) A logo representation of amino acid diversity among 8,168 C2H2-ZF proteins collected from zebrafish genes. (E) A histogram showing the A-site position of the leading ribosome in disome footprints relative to the C2H2-ZF sequence in zebrafish. (F) Disome occupancy at the A-, P-, or E-site codon of the leading stalled ribosome in all genes (x-axis) and C2H2 sequences (y-axis). *rho*, Spearman's rank correlation. (G) Violin plots showing the distribution of the predicted pI values in zebrafish genes. pI values of genes without C2H2-ZF (blue), genes with C2H2-ZF (turquoise), and C2H2-ZF genes up-regulated in MZ*znf598* (orange) are shown. (H) Mean net charge in the nascent chain around ribosome

collision sites on C2H2-ZF sequences in zebrafish. Nascent chain sequences determined by the A-site position of experimentally detected disome footprints (red) and those determined by the randomly chosen A-site position (blue) are shown. (I) Violin plots showing the distribution of the predicted mRNA stability by iCodon in zebrafish genes. The predicted mRNA stabilities of genes without C2H2-ZF (blue), genes with C2H2-ZF (turquoise), and C2H2-ZF genes up-regulated in MZ*znf598* (orange) are shown. The p values in G and I were calculated by the Wilcoxon rank sum test. The data underlying this figure can be found in S1 Data.
(TIF)

**S6 Fig. Analysis of BRIC-Seq and disome profiling data in human cells.** (A) A schematic of BRIC-Seq in HEK293T cells. (B) Cumulative distributions of fold changes in the mRNA half-life in HEK293T cells, comparing ZNF598 knockdown cells and control cells. All genes (black), C2H2-ZF genes (red), and RING-ZF genes (purple) are shown. The x-axis shows the fold changes in the mRNA half-life. The *p* values are shown on the left (Kolmogorov–Smirnov test). (C) A logo representation of amino acid diversity among C2H2-ZF collected from human genes. (D) A histogram showing the A-site position of the leading stalled ribosome in disome footprints relative to the C2H2-ZF sequence in HEK293 cells. (E) Mean net charge in the nascent chains around ribosome collision sites on C2H2-ZF sequences in HEK293 cells. Nascent chain sequences determined by the A-site position of experimentally detected disome footprints (red) and those determined by the randomly chosen A-site position (blue) are shown. The data underlying this figure can be found in S1 Data.
(TIF)

**S7 Fig. Validation of disome profiling and BRIC-Seq data in human cells.** (A–C) Distributions of disome footprints (blue) in the ZNF768, ZBTB7A, and ZNF100 ORFs. C2H2-ZF is indicated in orange. The fragments encoding C2H2-ZF repeats with high disome footprint amounts used for validation experiments are shown below with amino acid positions. (D) Results of the tandem ORF assay with human C2H2-ZF sequences in HEK293T cells. The graphs represent the average of 3 independent experiments. Relative Fluc signals normalized to Rluc signals are shown. Values with no insert (-) were set to one. The error bars show the standard deviation. The open circles show each data point. Black asterisks indicate p < 0.05 compared to no insert. Red asterisks indicate p < 0.05 compared to LacZ-N600 (Dunnett's test). (E) Time course qRT-PCR analysis of ZNF768, ZBTB7A, and ZNF100 mRNAs after Actinomycin D treatment in control (blue) and ZNF598 knockdown (red) HEK293T cells. The graphs represent the average of 3 independent experiments. The error bars show the standard deviation. (F) Western blotting to detect ZNF598 proteins in control and ZNF598 knockdown cells. β-actin was detected as a loading control. The data underlying this figure can be found in S1 Data.
(TIF)

**S1 Table. List of mRNAs upregulated in MZ*znf598* mutant at 6 hpf.**
(XLSX)

**S2 Table. List of primers.**
(XLSX)

**S1 Data. The underlying numerical data for main and supporting figures.**
(XLSX)

**S1 Raw Images. Raw Images of original gels and blots.**
(PDF)

## Acknowledgments

We thank our laboratory members for their discussions and critical comments on the project, Kaori Kaminoyama and Tomoaki Sakamoto for RNA-Seq, and Nozomi Ugajin for establishing the *rps10* K139/140R zebrafish strain. pColdI-GFP was a kind gift from Kotaro Tsuboyama and Yukihide Tomari. DNA libraries for ribosome profiling were sequenced by the Vincent J. Coates Genomics Sequencing Laboratory in QB3 Genomics at UC Berkeley (RRID: SCR_022170) at UC Berkeley (RRID:SCR_022170), supported by the National Institutes for Health (NIH) Instrumentation Grant (S10 OD018174). Computational analysis was supported by the supercomputer HOKUSAI Sailing Ship in RIKEN.

The Galaxy server that was used for some calculations funded in part by the Collaborative Research Centre 992 Medical Epigenetics (DFG grant SFB 992/1 2012) and the German Federal Ministry of Education and Research (BMBF grants 031 A538A/A538C RBC, 031L0101B/ 031L0101C de.NBI-epi, 031L0106 de.STAIR (de.NBI)).

## Author Contributions

**Conceptualization:** Yuichiro Mishima.

**Data curation:** Kimi Wakabayashi, Yuichiro Mishima.

**Formal analysis:** Kota Ishibashi, Yuichi Shichino, Peixun Han, Mari Mito, Seisuke Kimura, Shintaro Iwasaki, Yuichiro Mishima.

**Funding acquisition:** Kota Ishibashi, Yuichi Shichino, Seisuke Kimura, Shintaro Iwasaki, Yuichiro Mishima.

**Investigation:** Kota Ishibashi, Yuichi Shichino, Peixun Han, Kimi Wakabayashi, Mari Mito, Shintaro Iwasaki, Yuichiro Mishima.

**Methodology:** Kota Ishibashi, Yuichi Shichino, Peixun Han, Seisuke Kimura, Shintaro Iwasaki, Yuichiro Mishima.

**Project administration:** Yuichiro Mishima.

**Resources:** Toshifumi Inada, Seisuke Kimura, Yuichiro Mishima.

**Software:** Shintaro Iwasaki.

**Supervision:** Yuichiro Mishima.

**Validation:** Kota Ishibashi, Yuichi Shichino, Shintaro Iwasaki, Yuichiro Mishima.

**Visualization:** Yuichiro Mishima.

**Writing – original draft:** Yuichiro Mishima.

**Writing – review & editing:** Kota Ishibashi, Yuichi Shichino, Toshifumi Inada, Shintaro Iwasaki.

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
