## [Editor Report · Decision Letter 0]

8 Feb 2024

Dear Dr Mishima, 

Thank you for submitting your manuscript entitled "Translation of zinc finger domains induces ribosome collision and Znf598-dependent mRNA decay in vertebrates" for consideration as a Research Article by PLOS Biology. Please accept my apologies for the delay in getting back to you as we consulted with an academic editor about your submission.

Your manuscript has now been evaluated by the PLOS Biology editorial staff, as well as by an academic editor with relevant expertise, and I am writing to let you know that we would like to send your submission out for external peer review.

Once your full submission is complete, your paper will undergo a series of checks in preparation for peer review. After your manuscript has passed the checks it will be sent out for review. To provide the metadata for your submission, please Login to Editorial Manager (https://www.editorialmanager.com/pbiology) within two working days, i.e. by Feb 10 2024 11:59PM.

Kind regards,

Richard

Richard Hodge, PhD

rhodge@plos.org

PLOS

---

## [Decision Letter · Decision Letter 1]

24 Mar 2024

Dear Dr Mishima,

Thank you for your patience while your manuscript "Translation of zinc finger domains induces ribosome collision and Znf598-dependent mRNA decay in vertebrates" was peer-reviewed at PLOS Biology. Please accept my sincere apologies for the delays that you have experienced during the peer review process. Your manuscript has now been evaluated by the PLOS Biology editors, an Academic Editor with relevant expertise, and by three independent reviewers. 

In light of the reviews, which you will find at the end of this email, we would like to invite you to revise the work to thoroughly address the reviewers' reports.

As you will see, the reviewers are generally positive about your manuscript and think it is interesting and well done. However, they raise some overlapping concerns, including the overall strength of the data to support the claims that C2H2 domain-containing mRNAs are bona fide substrates of the no-go decay pathway and that they lead to ribosome stalling. Specifically, Reviewer #3 outlines several experiments to validate that the C2H2-ZF mRNAs undergo no-go decay in a Znf598 mutant background. In addition, Reviewer #1 also asks that few of the upregulated C2H2 candidate genes are validated in HEK cells as shown in zebrafish, as well as demonstrating similar trends of ribosome collision.

Given the extent of revision needed, we cannot make a decision about publication until we have seen the revised manuscript and your response to the reviewers' comments. Your revised manuscript is likely to be sent for further evaluation by all or a subset of the reviewers.

**IMPORTANT - SUBMITTING YOUR REVISION**

*Re-submission Checklist*

*Published Peer Review*

*PLOS Data Policy*

*Blot and Gel Data Policy*

Sincerely,

Richard

Richard Hodge, PhD

rhodge@plos.org

REVIEWS:

Reviewer #1: I find the manuscript titled "Translation of zinc finger domains induces ribosome collision and Znf598-dependent mRNA decay in vertebrates" well-crafted, presenting a clear hypothesis that addresses a significant research question within the realm of RNA decay in vertebrates. The study focuses on the role of the E3 ubiquitin ligase Znf598 in recognizing collided ribosomes on transcripts which codes for C2H2 zinc finger domain containing genes. This process is the part of ribosome-associated quality control (RQC) which eventually facilitates the rescue of stalled ribosomes and the degradation of stall-prone mRNAs by No Go Decay (NGD) mechanism.

The authors have meticulously selected the maternal zygotic Znf598 zebrafish mutant to address the role of znf598 during maternal zygotic transition, a phase known for the decay of maternal genes. Intriguingly, the study revealed a distinct set of differentially upregulated RNAs in these mutants that deviate from the well-established maternal and zygotic-controlled mRNA decay pathways. The validation of RNA-seq candidates through well-controlled qRT-PCR experiments which is followed by rescue experiments adds significant specificity to their findings.

Later they show zygotic candidates are densely distributed on chromosome 4 and with fish specific domain also these genes respond more strongly that other C2H2 domain containing genes. In later sets of experiments authors found the motif and sequences required for ribosome stalling collision in their well-executed reporter controls. Later using reporter assay they showed how ribosome collision leads to mRNA decay. Moreover, the authors replicated similar findings in human cell lines. Nonetheless, they cautiously conclude that determining the exact motif sequences required for ribosome stalling and degradation in both zebrafish and human cell lines poses significant challenges. 

Here are some specific minor comments and suggestions to improve the manuscript.

1. Maternal zygotic ZNF598 mutant doesn't show apparent early embryonic development phenotype, suggest this gene is not important during early embryonic development and around 5 dpf authors start seeing reduced body length in mutant larvae. How would you correlate differentially upregulated candidate's gene contribution by RNA-seq at 6hpf (no phenotype) to 5 dpf phenotype and again becoming like wild type fish at later developmental stages.

2. Authors have shown how C2H2 domain containing genes gets differentially upregulated in ZNF598 mutant background in zebrafish and they have nicely validated some of these candidates by QRT-PCR, rescue experiments, morpholinos by blocking the translation and finally by RNA decay experiments. They did similar kind of experiment in human HEK cell lines and claim that this process seems to be conserved among different vertebrates but if would be nice if they can validate few of these candidates in cell lines and show similar trends of ribosome collision followed by RNA decay. 

3. In almost all the experimental setup, authors have used proper controls and repeated experiments as required to run different statistical analysis, but it would be nice, if in figure 1 maternal zygotic ZNF598 zebrafish mutant experiments, authors should have sample size at least in double digits (N > 10). 

4. Figure2 C: In legend color name should be orange instead of pink. Also please check other colors name in different panels and in their respective legends. 

Reviewer #2: The manuscript by Ishibashi and colleagues explores the role of ribosome quality control (RQC) and no-go decay (NGD) processes in regulating the stability of maternal mRNAs in zebrafish during maternal-to-zygotic transition (MZT). The authors took advantage of the fact that both NGD and RQC critically depend on the presence of the E3 ligase Znf598, which adds ubiquitin chains to ribosomal proteins during collisions. In particular, they conducted RNAseq experiments on wild type and znf598 mutant strains at 1-hour and 6-hour post fertilization, and assessed how lack of the E3 ligase altered mRNA levels. Interestingly, they find that transcripts for genes encoding C2H2 zinc finger domains are enriched in the mutant strain relative to the wild type one. A number of these "NGD" target mRNAs were confirmed to be stabilized in mutant cells. The authors then went on to provide data that suggest that C2H2 domains induce ribosome stalling on dual luciferase reporters and assess the effect of their composition on translation elongation. Finally, they provide some evidence to show that the impact of C2H2 domains is general and induces NGD in human cells.

Overall, this was an interesting paper and provides new insights into the role of RQC (and Znf598-mediated sensing of stalled ribosomes) on gene regulation during animal development. The transcriptomic data appear very strong and support the conclusion that C2H2 domain are somewhat targeted for destabilization by NGD. However, there are some clarifications that could be done to improve the quality of the study. In particular, I felt that the argument that these sequences lead to ribosome stalling is not as strong as the transcriptomic data.

My general concerns are highlighted below:

1) The data in Figure 3E is confusing. Studies in yeast and mammalian cells have shown that deletion of Hel2/ZNF598 leads to readthrough of stall sequences. Yet, the data in this figure show the opposite. There is much less full-length product in the absence of Znf598, even though the mRNA is supposedly stabilized.

2) In the same figure, it seems that addition of any fragment from Znf236 results in some stalling. These are missing proper controls. Have the authors tried control sequences from unaffected mRNAs? They would have to be of similar lengths.

3) If I understood it correctly, the embryo development in the Znf598 mutant cells appears normal. If that is true, what is the biological implication of having C2H2-containing mRNA destabilized by RQC? If these are real targets, why do not they appear to be important?

4) I found some of the authors' arguments to be unjustified. For example, the statement "Comprehensive identification of the endogenous NGD target mRNAs is needed to understand the physiological role of NGD." By their nature, NGD and RQC are quality control processes that target aberrant RNAs, and the idea that they need to have endogenous targets is not accurate. Similarly, the statement "Chemically damaged mRNAs induce ribosome stalling and are potential NGD substrates; however, the frequency of such damage occurring on endogenous mRNAs is unclear" is unjustified. Cells evolved pathways to repair DNA and degrade damaged proteins, for example. The levels of damaged biological molecules might be low in a laboratory conditions, but under stress and in the presence of chemical agents, their levels are known to go up.

Reviewer #3: A new paradigm for no-go mRNA decay (NGD) has been discovered in eukaryotes: ribosome stalling leads to ribosome collisions and the formation of a new interface between the collided small ribosomal subunits. This new interface is recognized by an E3 ligase (Hel2 in yeast, Znf598 in vertebrates), which ubiquitinates ribosomal proteins at the interface: eS10 and uS10. Ubiquitination is a critical event that recruits factors required to cleave the mRNA (Cue2/NONU-1), split stalled ribosomes into subunits (ASCC), and ubiquitinate the nascent polypeptide to direct it toward elimination (Ltn1). Reporter mRNAs containing difficult-to-translate patches have been utilized in the field to study NGD and RQC; however, the endogenous mRNA substrates of these quality control pathways have remained poorly characterized. 

In this manuscript titled: "Translation of zinc finger domains induces ribosome collision and Znf598-dependent mRNA decay in vertebrates" by Ishibashi et al. employed in vivo zebrafish models to uncover that C2H2-ZF are endogenous mRNA substrates of NGD/RQC during the maternal-to-zygotic transition (MZT). Using RNA-seq, disome profiling, and reporter assays, the authors showed C2H2-ZF mRNAs induce ribosome collisions and subsequent degradation in a Znf598-dependent manner. Moreover, the authors showed the Znf598-mediated degradation of C2H2-ZF mRNA in human cells. 

The authors have made an intriguing discovery and data presented in the manuscript are sound. However, critical experiments are missing to fully support their conclusions. The manuscript would be suitable for this journal after addressing the specific points below. 

Specific points:

1. The authors have shown that C2H2-ZF mRNA are stall- and collision-prone and degraded in a Znf598-dependent manner. RNA-seq analysis at different stages revealed significant changes of many genes in MZznf598 (Figure 1D), likely reflecting the role of Znf598 in other biological processes and/or secondary effects. Direct evidence for C2H2-ZF mRNA being NGD substrates is missing. The author should test whether ablation of the conserved homolog of Cue2/NONU-1, N4BP2, can rescue the abundance of these endogenous mRNAs and the reporter mRNAs. In addition, a hallmark of NGD is endonucleolytic cleavage. The authors should also examine whether decay intermediates could be captured using their reporters. 

2. Complementary to Specific point 1 above, a key feature of RQC is CAT-tailing. If these C2H2-ZF are endogenous substrates of RQC, they should undergo CAT-tailing. It may be challenging to monitor some of these endogenous proteins by immunoblotting due to their low abundance. However, the authors have established a reporter system using these stall-prone sequences. Such a setup could be used to examine CAT-tailing.

---

## [Decision Letter · Decision Letter 2]

23 Sep 2024

Dear Dr Mishima,

Thank you for your patience while we considered your revised manuscript "Translation of zinc finger domains induces ribosome collision and Znf598-dependent mRNA decay in zebrafish" for publication as a Research Article at PLOS Biology. This revised version of your manuscript has been evaluated by the PLOS Biology editors, the Academic Editor and the original reviewers.

Based on the reviews, I am pleased to say that we are likely to accept this manuscript for publication, provided you satisfactorily address the following data and other policy-related requests that I have provided below (A-F):

(A) Please include the full name of the IACUC/ethics committee that specifically reviewed and approved the zebrafish studies. Please also include an approval number. 

(B) You may be aware of the PLOS Data Policy, which requires that all data be made available without restriction: http://journals.plos.org/plosbiology/s/data-availability. For more information, please also see this editorial: http://dx.doi.org/10.1371/journal.pbio.1001797

-Supplementary files (e.g., excel). Please ensure that all data files are uploaded as 'Supporting Information' and are invariably referred to (in the manuscript, figure legends, and the Description field when uploading your files) using the following format verbatim: S1 Data, S2 Data, etc. Multiple panels of a single or even several figures can be included as multiple sheets in one excel file that is saved using exactly the following convention: S1_Data.xlsx (using an underscore).

-Deposition in a publicly available repository. Please also provide the accession code or a reviewer link so that we may view your data before publication. 

Figure 1D, 2B-C, 2E-G, 3A, 3C, 3E, 3G, 4A-B, 4D-F, 5C-D, 5G-H, 5J, 6B-C, 6E, S1A-B, S1D-E, S2A-E, S3A-D, S4A-B, S5A-I, S6A-E, S7D-E 

(C) Thank you for providing the RNA-seq data in the GEO database (GSE236143, GSE236144 and GSE233374). However, we note that the data for GSE236143 and GSE236144 are currently on hold for release. We ask that you please make all sequencing data publicly available at this stage before publication.

(D) Please also ensure that each of the relevant figure legends in your manuscript include information on *WHERE THE UNDERLYING DATA CAN BE FOUND*, and ensure your supplemental data file/s has a legend.

(E) Please ensure that your Data Statement in the submission system accurately describes where your data can be found and is in final format, as it will be published as written there. 

(F) Per journal policy, if you have generated any custom code during the course of this investigation, please make it available without restrictions. Please ensure that the code is sufficiently well documented and reusable, and that your Data Statement in the Editorial Manager submission system accurately describes where your code can be found. 

We expect to receive your revised manuscript within two weeks. 

*Published Peer Review History*

*Press*

Kind regards,

Richard

Richard Hodge, PhD

rhodge@plos.org

Reviewer remarks:

Reviewer #1: 

1. The authors have addressed most of the reviewers' suggestions appropriately. In some instances where they were unable to provide sufficient evidence, they have toned down their statements and in certain cases moved those results to supplementary section, acknowledging the limitations of their findings.

2. I acknowledge the limitations of working with animal models, where smaller sample sizes can impact the robustness of conclusions. However, for the main figures, the authors have sufficiently increased the sample size to support their findings. In cases where they were unable to do so for the supplementary figures, they have recognized these limitations and appropriately moderated their statements to conclude their findings. 

3. Since Authors could not fully recapitulate zebrafish data in human cell lines experiments, therefore they have moved this part to supplementary figure sections citing that still this human cell line data will help other researchers to compare results between zebrafish and human cells. I agree with authors, and I found it suitable to be the part of the supplementary section. 

4. Authors have made appreciate changes to fix inconsistencies with colors and typos. 

Overall, I believe, authors have satisfactorily made most of the suggested changes in the revised manuscript and this manuscript has the potential to make a valuable contribution to PLOS Biology. I appreciate the effort invested by the authors in this research and look forward to seeing the final version of the manuscript.

Reviewer #2: The authors addressed my main concerns, and in particular I appreciate that they tuned down the interpretation of the human data. The revised manuscript provides new insights into the role of ribosome quality control in regulating gene expression in zebrafish. I am supportive of publication. 

Reviewer #3: The authors have significantly improved the manuscript. Although some experiments did not yield anticipated results, I remain enthusiastic about the work. This study will lay the groundwork for future research on RQC and NGD in vertebrates.

---

## [Editor Report · Decision Letter 3]

7 Oct 2024

Dear Dr Mishima,

On behalf of my colleagues and the Academic Editor, Wendy Gilbert, I am pleased to say that we can accept your manuscript for publication, provided you address any remaining formatting and reporting issues. These will be detailed in an email you should receive within 2-3 business days from our colleagues in the journal operations team; no action is required from you until then. Please note that we will not be able to formally accept your manuscript and schedule it for publication until you have completed any requested changes.

PRESS

Best wishes, 

Richard

Richard Hodge, PhD

rhodge@plos.org

PLOS
